# Pharmacological augmentation of nicotinamide phosphoribosyltransferase (NAMPT) protects against paclitaxel-induced peripheral neuropathy

Peter M LoCoco, April L Risinger, Hudson R Smith, Teresa S Chavera, Kelly A Berg, William P Clarke*

Department of Pharmacology, University of Texas Health Science Center at San Antonio, San Antonio, United States

**Abstract** Chemotherapy-induced peripheral neuropathy (CIPN) arises from collateral damage to peripheral afferent sensory neurons by anticancer pharmacotherapy, leading to debilitating neuropathic pain. No effective treatment for CIPN exists, short of dose-reduction which worsens cancer prognosis. Here, we report that stimulation of nicotinamide phosphoribosyltransferase (NAMPT) produced robust neuroprotection in an aggressive CIPN model utilizing the frontline anticancer drug, paclitaxel (PTX). Daily treatment of rats with the first-in-class NAMPT stimulator, P7C3-A20, prevented behavioral and histologic indicators of peripheral neuropathy, stimulated tissue NAD recovery, improved general health, and abolished attrition produced by a near maximum-tolerated dose of PTX. Inhibition of NAMPT blocked P7C3-A20-mediated neuroprotection, whereas supplementation with the NAMPT substrate, nicotinamide, potentiated a subthreshold dose of P7C3-A20 to full efficacy. Importantly, P7C3-A20 blocked PTX-induced allodynia in tumored mice without reducing antitumoral efficacy. These findings identify enhancement of NAMPT activity as a promising new therapeutic strategy to protect against anticancer drug-induced peripheral neurotoxicity.

DOI: https://doi.org/10.7554/eLife.29626.001

*For correspondence:
clarkew@uthscsa.edu

## Introduction

The microtubule stabilizer, paclitaxel (PTX), is widely used for the treatment of breast, ovarian, lung, and pancreatic cancer (*Rivera and Cianfrocca, 2015*). Despite its clinical effectiveness, however, PTX often produces debilitating, dose-limiting peripheral neuropathy. Chemotherapy-induced peripheral neuropathy (CIPN) is the most common nonhematologic side effect of anticancer pharmacotherapy that affects up to 90% of cancer patients (*Grisold et al., 2012*; *Miltenburg and Boogerd, 2014*; *Windebank and Grisold, 2008*). The damage to peripheral sensory neurons resulting from anticancer treatment causes patients to experience stimulus-specific allodynia (i.e. pain in response to innocuous stimuli), tingling pain, numbness, and/or loss of sensory function, that are symmetrically-distributed largely in their hands and feet (*Speck et al., 2013*). The pain associated with this neuropathy intensifies with each cycle of chemotherapy and persists beyond the cancer treatment period, often indefinitely, subjugating patients to a substandard quality of life both during and after treatment (*Cavaletti and Marmiroli, 2010*). As there are no effective treatments or preventions for CIPN, potentially life-saving cancer treatment must often be dose-reduced or discontinued, adversely affecting cancer prognosis and survival (*Rivera and Cianfrocca, 2015*). Consequently, there is an urgent need for approaches to prevent CIPN and thereby improve both cancer treatment and the quality of life of cancer patients.

Previously, an in vivo phenotypic screen searching for proneurogenic compounds revealed that the aminopropyl carbazole, P7C3, increased the number of newborn neurons in the mouse hippocampus (*Pieper et al., 2010*). Subsequent experiments demonstrated that rather than increasing neurogenesis, P7C3 reduced apoptosis in differentiating neuronal progenitors, suggesting a neuroprotective effect that promoted survival of the newborn neurons. P7C3-A20, a structural analogue of P7C3, displayed strong neuroprotective efficacy in several models of neurodegeneration, including Parkinson's disease, amyotrophic lateral sclerosis, traumatic brain injury, optic nerve injury, ischemic stroke, and sciatic nerve crush (*De Jesús-Cortés et al., 2012*; *Kemp et al., 2015*; *Loris et al., 2017*; *Oku et al., 2017*; *Tesla et al., 2012*; *Wang et al., 2016a*; *Yin et al., 2014*). Recent work identified P7C3-A20 as a first-in-class stimulator of nicotinamide phosphoribosyltransferase (NAMPT), the rate-limiting enzyme in the nicotinamide adenine nucleotide (NAD) salvage pathway (*Wang et al., 2014*). Here, using an aggressive rodent model of CIPN, we report that P7C3-A20 protected peripheral sensory neurons from neurotoxicity induced by PTX and that this protection required NAMPT activity. Importantly, P7C3-A20 did not interfere with the antitumor activity of PTX nor promote tumor growth. These results suggest that enhancement of NAMPT activity with P7C3-A20 may be a promising new therapeutic strategy to protect peripheral afferent sensory neurons against anticancer drug-induced peripheral neurotoxicity.

## Results

### Aggressive PTX treatment produces peripheral neuropathy and damages peripheral afferent neurons

To induce peripheral neuropathy, rats were treated with a near maximum-tolerated dose of PTX. Adult male Sprague-Dawley rats received three injections of PTX (11.7 mg/kg/day, i.p.), administered every other day, for a total cumulative dose of 35 mg/kg (*Figure 1—figure supplement 1A*). As is typical with this dose of PTX (*Cliffer et al., 1998*), average body weights and circulating leukocytes maximally decreased by 16% and 60%, respectively, following which the animals began to recover (*Figure 1—figure supplement 1B and C*). Altered nociceptive thresholds to mechanical, thermal cold, and thermal heat stimulation are robust indicators of the development of peripheral neuropathy and commonly observed in patients with CIPN (*Argyriou et al., 2012*; *Dougherty et al., 2007*; *Dougherty et al., 2004*; *Kleggetveit et al., 2012*). PTX treatment of rats significantly reduced the thresholds for mechanical and cold stimuli to elicit a paw withdrawal response. This increased sensitivity (allodynia) developed within 4 days and was sustained for over 3 weeks (*Figure 1A–F*). By contrast, PTX-treated rats developed a transient hypoalgesia (reduced sensitivity) to thermal heat stimulation (*Figure 1G–I*). Differential sensitivities to external stimuli have been described in cancer patients receiving PTX (*Cata et al., 2006*; *Dougherty et al., 2004*; *Nahman-Averbuch et al., 2011*) as well as in rodent models that incorporate moderate to high cumulative PTX dosages (*Authier et al., 2000*; *Peters et al., 2007b*).

Degeneration of intraepidermal nerve fibers (IENFs), the tortuous free nerve endings of nociceptive neurons that innervate the epidermal layer of peripheral tissues, is a signature of PTX-induced damage to peripheral nociceptive neurons (*Jin et al., 2008*; *Krukowski et al., 2015*; *Liu et al., 2010*; *Siau et al., 2006*). PTX treatment significantly reduced IENF density by ~50% in biopsies from rat hindpaws and forepaws obtained on day 7, 3 days after the final PTX injection (*Figure 1J and L*). IENF degeneration was still evident more than 2 weeks later (*Figure 1M*), which also paralleled the persistent allodynia observed in the rats. We extended our histological analysis to include measurement of the neuronal injury marker, activating transcription factor 3 (ATF3) in perikarya of lumbar dorsal root ganglia (DRG). ATF3 is up-regulated in peripheral and spinal neurons following neuronal injury (e.g. axotomy) or stress (*Tsujino et al., 2000*). Moderate-to-high doses of PTX have been shown to induce ATF3 expression in rat DRG neurons (*Liu et al., 2010*; *Peters et al., 2007a*; *Verheyen et al., 2012*). PTX treatment produced a marked increase in the number of lumbar DRG neurons expressing ATF3 within 3 days after treatment (*Figure 1K and N*).

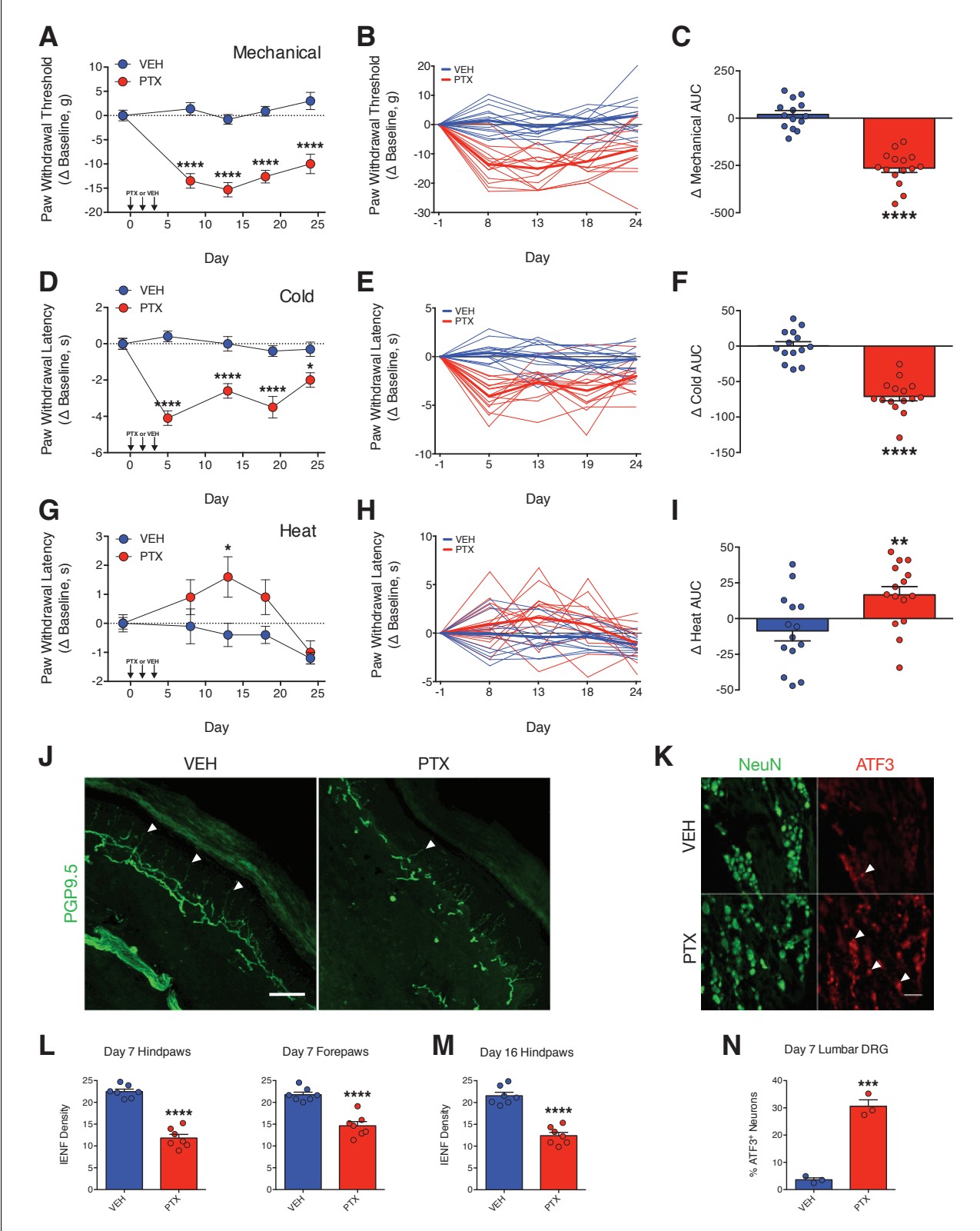

**Figure 1.** PTX differentially affects nociceptive thresholds and damages peripheral sensory neurons. (A–I) Nociceptive thresholds to mechanical (A), cold (D), and heat (G) stimulation of the hindpaws of adult male Sprague-Dawley rats treated with vehicle (EtOH/Kolliphor EL/PBS, 1:1:6, i.p.) or PTX (11.7 mg/kg, i.p.) on days 0, 2, and 4. Data represent the mean change from baseline ± SEM. Individual rat timecourse plots showing changes in mechanical (B), cold (E), or heat (H) sensitivity following vehicle or PTX treatment. Bold lines represent group means. Area under the timecourse curves

*Figure 1 continued on next page*

*Figure 1 continued*

(AUC) of mechanical (**C**), cold (**F**), or heat (**I**) thresholds from vehicle- or PTX-treated rats. Bars represent the mean ± SEM and small circles are individual rat AUC values. ****p<0.0001, **p<0.01, *p<0.05 vs. Veh by two-way mixed effect ANOVA with Sidak's post-hoc test (**A, D, G**) or two-tailed Student's t-test (**C, F, I**), n = 14–15 rats/group. (**J**) Confocal images of IENFs in rat hindpaw biopsies. IENFs immunolabeled with PGP9.5 (arrowheads) project from subepidermal fascicles across the epidermal-dermal junction. Scale bar, 100 μm. (**K**) Confocal images of ATF3 expression (arrowheads) in nuclei of DRG neurons (also labeled with NeuN) acquired on day 7. Scale bar, 100 μm. (**L and M**) Quantification of IENF densities from paw biopsies collected on day 7 (**L**) and day 16 (**M**) of the experimental paradigm. Bars represent the mean ± SEM calculated from individual rat IENF densities from each group (circles), n = 7 rats/group. (**N**) Quantification of ATF3 expression. Bars represents mean ± SEM from each group, n = 3 rats/group. ****p<0.0001, ***p<0.001 vs. Veh by two-tailed Student's t-test (**L–N**).

DOI: https://doi.org/10.7554/eLife.29626.002

The following figure supplement is available for figure 1:

**Figure supplement 1.** PTX treatment produces a recoverable loss in both body weight and circulating leukocyte levels in adult male rats.

DOI: https://doi.org/10.7554/eLife.29626.003

## P7C3-A20 abrogates neuropathic pain, protects peripheral nociceptive neurons from damage, improves general health, and reduces attrition associated with PTX treatment

To test the hypothesis that P7C3-A20 would prevent the development of peripheral neuropathy, rats were treated with P7C3-A20 (10 mg/kg/day, i.p.) beginning 2 days prior to the first injection of PTX and continuing daily until the conclusion of the paradigm (*Figure 2—figure supplement 1A*). P7C3-A20 treatment did not alter PTX-induced weight loss or leukopenia (*Figure 2—figure supplement 1B and C*), suggesting no protection for rapidly dividing gastrointestinal epithelia or leukocytes. Remarkably, however, P7C3-A20 prevented completely the development of PTX-induced mechanical allodynia and heat hypoalgesia, and markedly reduced cold allodynia (*Figure 2A–F*). P7C3-A20 alone had no effect on nociceptive responses as compared to vehicle, indicating its action was not analgesic in nature.

In concordance with the results from the behavior experiments, P7C3-A20 completely prevented PTX-induced IENF degeneration by day 7, and this protective effect was sustained through at least day 16 (*Figure 2G and I*). P7C3-A20 also partially reduced PTX-induced ATF3 expression from 33.4% to 20.6% of lumbar DRG neurons (*Figure 2H and J*). It is noteworthy that P7C3-A20 did not completely prevent the up-regulation of ATF3 by PTX treatment, suggesting that, even in the presence of P7C3-A20, PTX still initiated a neuronal damage/stress response. Nevertheless, the histological analyses clearly indicate that P7C3-A20 protected peripheral afferent neurons from PTX-induced damage.

To confirm our initial findings that P7C3-A20 abrogated PTX-induced neuropathy, we conducted a randomized, double-blinded dose-response study of P7C3-A20 and a second analogue, P7C3-S321. As before, P7C3-A20 reduced the PTX-induced mechanical and cold allodynia as well as IENF degeneration in a dose-dependent manner and consistent with plasma drug levels for each (*Figure 3A–C* and *Figure 3—figure supplements 1* and *2*, *Figure 3—source data 1*). P7C3-S321 was partially effective at preventing PTX-induced mechanical allodynia but did not reduce the cold allodynia. Changes in nociceptive thresholds to mechanical and cold stimulation were strongly correlated with IENF density (*Figure 3D and E*, *Figure 3—source data 1*). Despite the dramatic neuroprotective effect on PTX-induced allodynia and IENF density, neither P7C3-A20 nor P7C3-S321 altered PTX-induced weight loss, although the highest dose of P7C3-A20 (20 mg/kg/day) partially reduced leukopenia (*Figure 3—figure supplement 3*, *Figure 3—source data 1*). Observationally, P7C3-A20, more so than P7C3-S321, improved indices of general animal health (*Table 1* and *Figure 3—figure supplement 4*). Furthermore, P7C3-A20 substantially reduced attrition rates across all behavioral experiments (*Figure 3F*). Study attrition was 25% for rats treated with PTX alone, where death almost always occurred within days 8 and 11 of the experimental paradigm (*Figure 3G*). In stark contrast, no deaths occurred in rats treated with P7C3-A20 at doses of at least 6.6 mg/kg/day.

We next compared the neuroprotective efficacy of P7C3-A20 with that of the inhibitor of poly (ADP)-ribose (PAR) polymerase (PARP), A-861696 (*Figure 4—figure supplement 1A*). In addition to anticancer activity, PARP inhibitors are under clinical investigation for efficacy to prevent CIPN (*Ramalingam et al., 2017*). A-861696, as well as its enantiomer and current clinical candidate, veliparib, were reported previously to attenuate mechanical hypersensitivity in rodents following

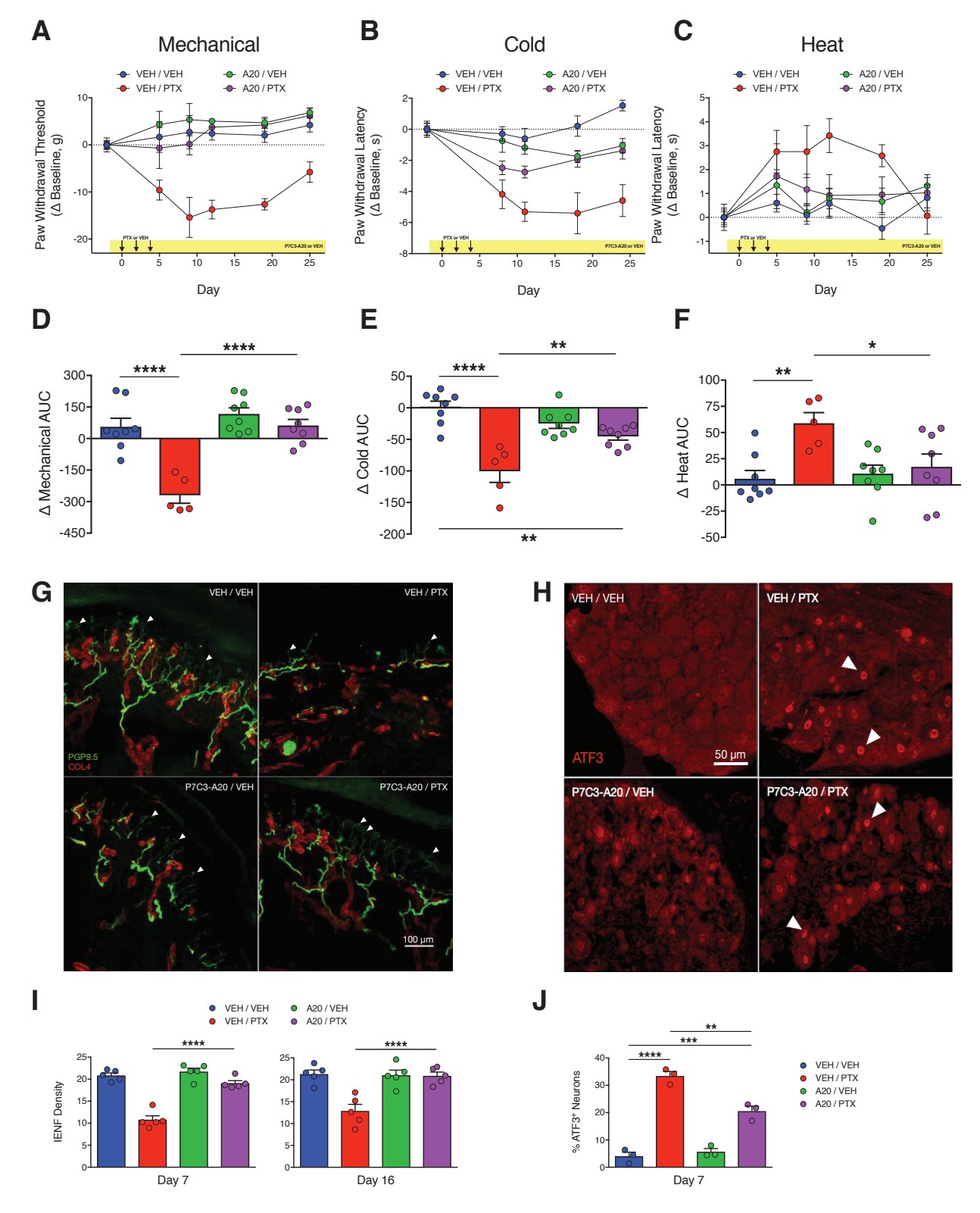

**Figure 2.** P7C3-A20 abrogates neuropathic pain and protects peripheral sensory neurons from PTX-induced damage. (A–C) Nociceptive thresholds to mechanical (A), cold (B), and heat (C) stimulation of the hindpaws of rats treated daily with vehicle (DMSO/Kolliphor EL/PBS, 1:4:10, i.p.) or P7C3-A20 (10 mg/kg/day, i.p.), in addition to vehicle or PTX treatment on days 0, 2, and 4 as before (*Figure 1*). Data represent the mean change from baseline ± SEM. (D–F) Treatment group AUCs of mechanical (D), cold (E), or heat (F) thresholds. Bars represent the mean AUC ±SEM and small circles

*Figure 2 continued on next page*

*Figure 2 continued*

are individual rat AUC values. ****p<0.0001, **p<0.01, *p<0.05 by one-way ANOVA with Dunnett's post-hoc test, n = 5–8 rats/group. (G and H) Confocal images of IENFs in rat hindpaw biopsies (G, scale bar: 100 μm) and ATF3 expression in lumbar DRG (H, scale bar: 50 μm.) (I) IENF densities from hindpaw biopsies collected on days 7 and 16 of the experimental paradigm. Bars represent the mean ±SEM calculated from individual rat IENF densities from each group (circles), n = 5–8 rats/group. (J) ATF3 expression in lumbar DRG perikarya collected on day 7. Bars represents mean ± SEM from each group, n = 3 rats/group. ****p<0.0001, **p<0.01 vs. Veh/PTX by one-way ANOVA with Tukey's post-hoc test (I and J).
DOI: https://doi.org/10.7554/eLife.29626.004

The following figure supplement is available for figure 2:

**Figure supplement 1.** Effects of P7C3-A20 on PTX-induced weight loss and leukopenia.
DOI: https://doi.org/10.7554/eLife.29626.005

treatment with vincristine or cisplatin (*Brederson et al., 2012*; *Ta et al., 2013*). In contrast to the robust neuroprotective efficacy of P7C3-A20, daily injections of A-861696 (50 mg/kg, i.p.) did not prevent the development of mechanical allodynia, nor did it prevent IENF degeneration triggered by a near maximum tolerated dose of PTX (*Figure 4A and B* and *Figure 4—figure supplement 1B–D*). We confirmed that A-861696 inhibited PARP in vivo as indicated by reduced PAR accumulation in lumbar DRG compared to PTX-only controls (*Figure 4C and D*). The ineffectiveness of A-861696 here, as compared with previous preclinical studies (*Brederson et al., 2012*; *Ta et al., 2013*), may due to the severity of peripheral neurotoxicity produced by the aggressive dose of PTX used in our model. Regardless, in a head-to-head comparison, P7C3-A20 demonstrated markedly superior efficacy compared with PARP inhibition in this CIPN model system.

## Augmentation of NAMPT activity is required for P7C3-A20-mediated neuroprotection

P7C3-A20 was reported to be a first-in-class stimulator of NAMPT, the rate-limiting enzyme in the NAD salvage pathway (*Wang et al., 2014*). If enhancement of NAD salvage is the mechanism by which P7C3-A20 protects peripheral sensory neurons from PTX-induced damage, we hypothesized that inhibition of NAMPT would prevent P7C3-A20-mediated neuroprotection. Global knockout of NAMPT is embryonic lethal and heterozygotes still express functional NAMPT (*Revollo et al., 2007*), therefore we opted to antagonize NAMPT with the selective NAMPT inhibitor, FK866 (*Hasmann and Schemainda, 2003*). To inhibit NAMPT in vivo, we utilized a twice-a-day dosing paradigm of FK866 that was reported previously to be devoid of toxicity and only slightly reduced tissue NAD levels (*Song et al., 2014*). Rats were treated twice daily with either FK866 (0.5 mg/kg, i.p., b.i.d.) or vehicle, along with P7C3-A20 (10 mg/kg/day) or vehicle starting on day −2. PTX was administered on days 0, 2, and 4 as before (*Figure 5A* and *Figure 5—figure supplement 1A*). FK866 alone did not alter body weight, mechanical sensitivity, or IENF density, nor did it exacerbate mechanical allodynia or IENF loss induced by PTX treatment. However, FK886 completely blocked the protective effects of P7C3-A20 on PTX-induced mechanical allodynia and IENF degeneration (*Figure 5B and C* and *Figure 5—figure supplement 1C*), suggesting that NAMPT activity is required for the neuroprotective effect of P7C3-A20 on PTX-induced peripheral neuropathy.

To further test the hypothesis that the neuroprotective effect of P7C3-A20 is dependent on its ability to enhance NAMPT activity, we evaluated the effect of supplementation of an ineffective dose of P7C3-A20 with the NAMPT substrate, nicotinamide (NAM), on PTX-induced allodynia and IENF degeneration (*Figure 5D* and *Figure 5—figure supplement 2A*). We elected to use a low daily dose of NAM (150 mg/kg/day, s.c.) that is not associated with neuroprotective efficacy (*Feng et al., 2006*; *Stevens et al., 2007*). Neither a subthreshold dose of P7C3-A20 (2.2 mg/kg/day) nor NAM alone altered PTX-induced reductions in body weight, mechanical threshold, or IENF density. However, NAM supplementation enhanced the neuroprotective efficacy of subthreshold P7C3-A20 (2.2 mg/kg/day) in both behavioral and anatomical indices of neuropathy to a degree equivalent to that produced by the highest dose of P7C3-A20 (20 mg/kg/day) (*Figure 5E and F* and *Figure 5—figure supplement 2B and C*). These results taken together reveal a critical contribution of NAMPT activity to the neuroprotective efficacy of P7C3-A20 against PTX-induced neuronal damage and peripheral neuropathic pain.

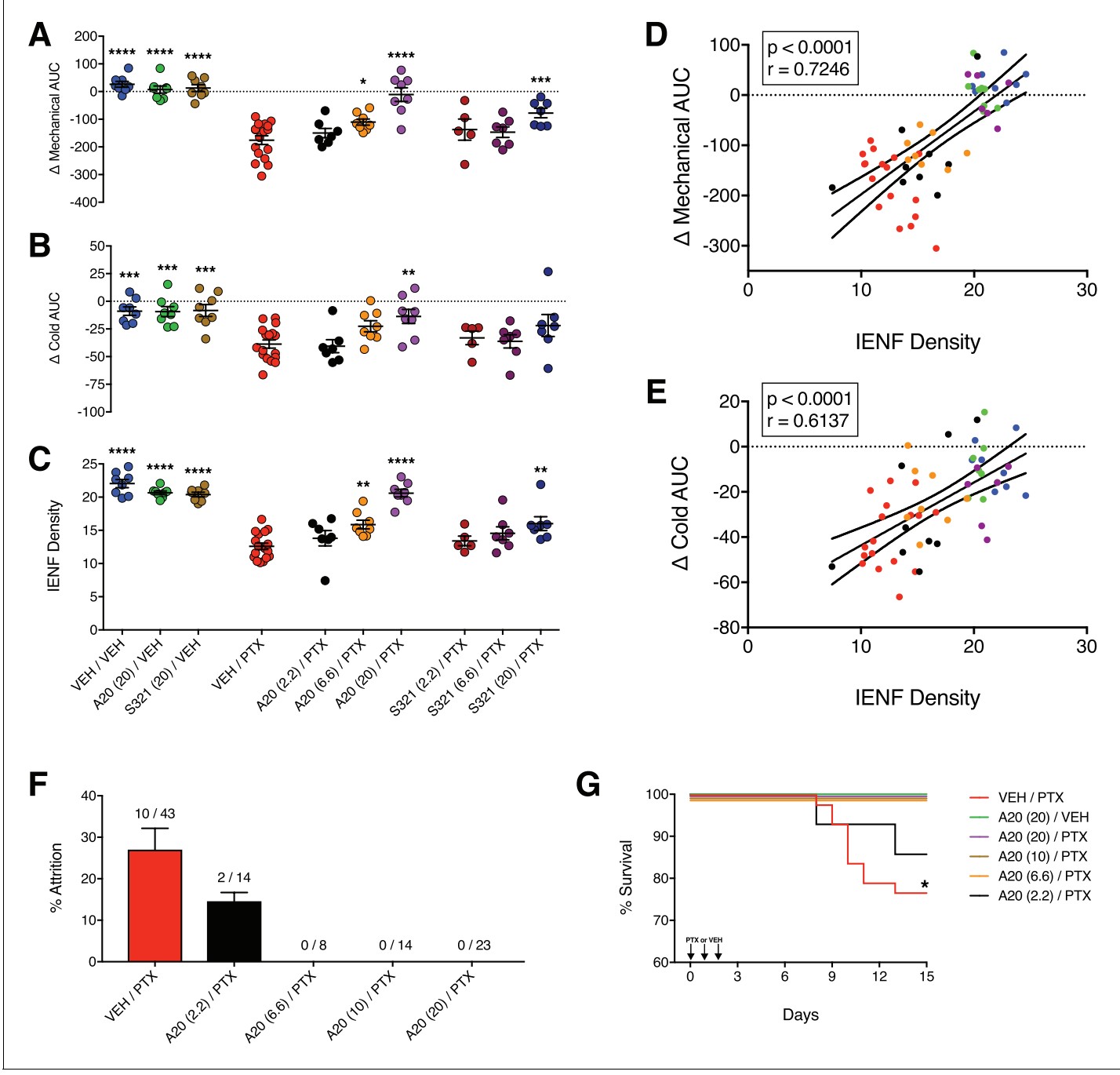

**Figure 3.** P7C3-A20 and P7C3-S321 are dose-dependently neuroprotective, improve general health, and attenuate premature death associated with PTX. (A–C) AUCs to mechanical (A) and cold (B) stimulation and IENF densities (C) showing the dose-dependent neuroprotective effects of P7C3-A20 and P7C3-S321. Horizontal lines represent mean AUC ±SEM calculated from individual rat AUC values shown as small circles. ****p<0.0001, ***p<0.001, **p<0.01, *p<0.05 vs. Veh/PTX by one-way ANOVA with Dunnett's post-hoc test, n = 5–17 rats/group. (D and E). Correlation analyses between individual rat IENF density and their respective mechanical (D) or cold (E) AUC (Pearson, two-tailed, p<0.0001). Black lines are linear regression curves with 95% confidence bands. Colors reflect treatment group as defined in **Figure 3A–C**. (F) Study attrition by treatment group. For each behavioral experiment, the number of rats removed due to >20% wt loss or death was divided by the total number of rats per treatment group. Data represent mean ± SEM, n = 1–4 independent experiments. (G) Survival curves showing attrition of rats treated only with PTX (red line) typically occurred between days 8–11, which was abolished by P7C3-A20 treatment. *p=0.0206 ($\chi^2$=13.32) by the Mantel-Cox log-rank test.

DOI: https://doi.org/10.7554/eLife.29626.006

The following source data and figure supplements are available for figure 3:

**Source data 1.** Raw datasets for **Figure 3** and all figure supplements.

*Figure 3 continued on next page*

*Figure 3 continued*

DOI: https://doi.org/10.7554/eLife.29626.011

**Figure supplement 1.** Dose-dependent effects of P7C3-A20 and P7C3-S321 on PTX-induced mechanical and cold allodynia.

DOI: https://doi.org/10.7554/eLife.29626.007

**Figure supplement 2.** Body weight, leukocyte counts, mechanical thresholds, and cold thresholds of all individual rats treated with vehicle or P7C3-A20 ± PTX.

DOI: https://doi.org/10.7554/eLife.29626.008

**Figure supplement 3.** Effects of P7C3-A20 and P7C3-S321 on PTX-induced weight loss and leukopenia.

DOI: https://doi.org/10.7554/eLife.29626.009

**Figure supplement 4.** P7C3-A20 improved general health of PTX-treated rats.

DOI: https://doi.org/10.7554/eLife.29626.010

## P7C3-A20 enhances NAMPT-mediated NAD recovery in response to cellular damage

To assess the ability of P7C3-A20 to stimulate NAMPT activity in neuronal cells, we treated A1A1 rat cortical neurons (*Berg et al., 1994*) with vehicle (ddH$_2$O) or H$_2$O$_2$ (200 µM) for 30 min, followed by treatment with P7C3-A20 (0.03–3 µM) or NAM (1 mM) and measured NAD levels. Treatment with H$_2$O$_2$ reduced NAD levels by 25%. While P7C3-A20 or NAM did not increase NAD production in vehicle-treated cells (*Figure 6A*), P7C3-A20 dose-dependently increased H$_2$O$_2$-depleted NAD levels back to vehicle-treated baseline (*Figure 6B*). The rescue of NAD levels in H$_2$O$_2$-treated cells was blocked by FK866 (*Figure 6C*). These data suggest that P7C3-A20 stimulates NAD recovery through NAMPT in neuronal cells, but this effect occurred only in cells depleted of NAD.

Next, we determined if P7C3-A20 could stimulate NAD in peripheral sensory neurons in vivo in response to PTX-induced neurotoxic damage. Rats were treated with P7C3-A20 (or vehicle) and PTX (or vehicle) according to our standard protocol (see *Figure 4—figure supplement 1*). NAD$^+$ was extracted from homogenized glabrous hindpaw skin, sciatic nerve, and lumbar DRG (collected on day 10 following the start of PTX treatment). These tissues were chosen because they represent anatomically distinct regions of peripheral afferent fibers, each with unique histologic characteristics (e.g. vascular permeability) that affect PTX accumulation in the tissue (*Abram et al., 2006*; *Hirakawa et al., 2004*; *Jimenez-Andrade et al., 2008*; *Olsson, 1968*). PTX treatment alone reduced

**Table 1.** Summary of general health observations in rats treated with P7C3-A20, P7C3-S321, or vehicle ±PTX

| General health observations | VEH/ VEH [n = 8] | VEH/PTX [n = 20] | A20 (20)/ VEH [n = 8] | A20 (20)/ PTX [n = 8] | A20 (6.6)/ PTX [n = 8] | A20 (2.2)/ PTX [n = 8] | S321 (20)/ VEH [n = 8] | S321 (20)/ PTX [n = 8] | S321 (6.6)/ PTX [n = 8] | S321 (2.2)/ PTX [n = 8] |
|---|---|---|---|---|---|---|---|---|---|---|
| Gnawing on observation box | - | 7 | - | - | 1 | 2 | - | 1 | - | - |
| Diarrhea | - | 9 | - | 2 | 4 | 5 | - | 4 | 4 | 4 |
| Mild hair loss | - | 14 | - | 3 | 5 | 6 | - | 5 | 6 | 6 |
| Hind limb paralysis | - | - | - | - | - | - | - | - | - | - |
| Chromo-dacryorrhea | - | 10 | - | - | - | 1 | - | 1 | 1 | 3 |
| Standing/walking on 'Tip-Toes' | - | 17 | - | 2 | 4 | 5 | - | 2 | 5 | 5 |
| Hovering paw | - | 4 | - | - | 1 | 2 | - | - | - | 1 |
| Jump response | - | 5 | - | 1 | 2 | 2 | - | 2 | 2 | 1 |
| Death | - | 3 | - | - | - | 1 | - | 1 | 1 | 3 |

Numbers in parentheses next to the drug name indicate the daily dose in mg/kg.

Numbers in the table represent the number of rats that displayed the health indicator in the row title. "- "indicates 0 rats displayed the health indicator. Total number of rats in the group are shown in the column title row.

DOI: https://doi.org/10.7554/eLife.29626.012

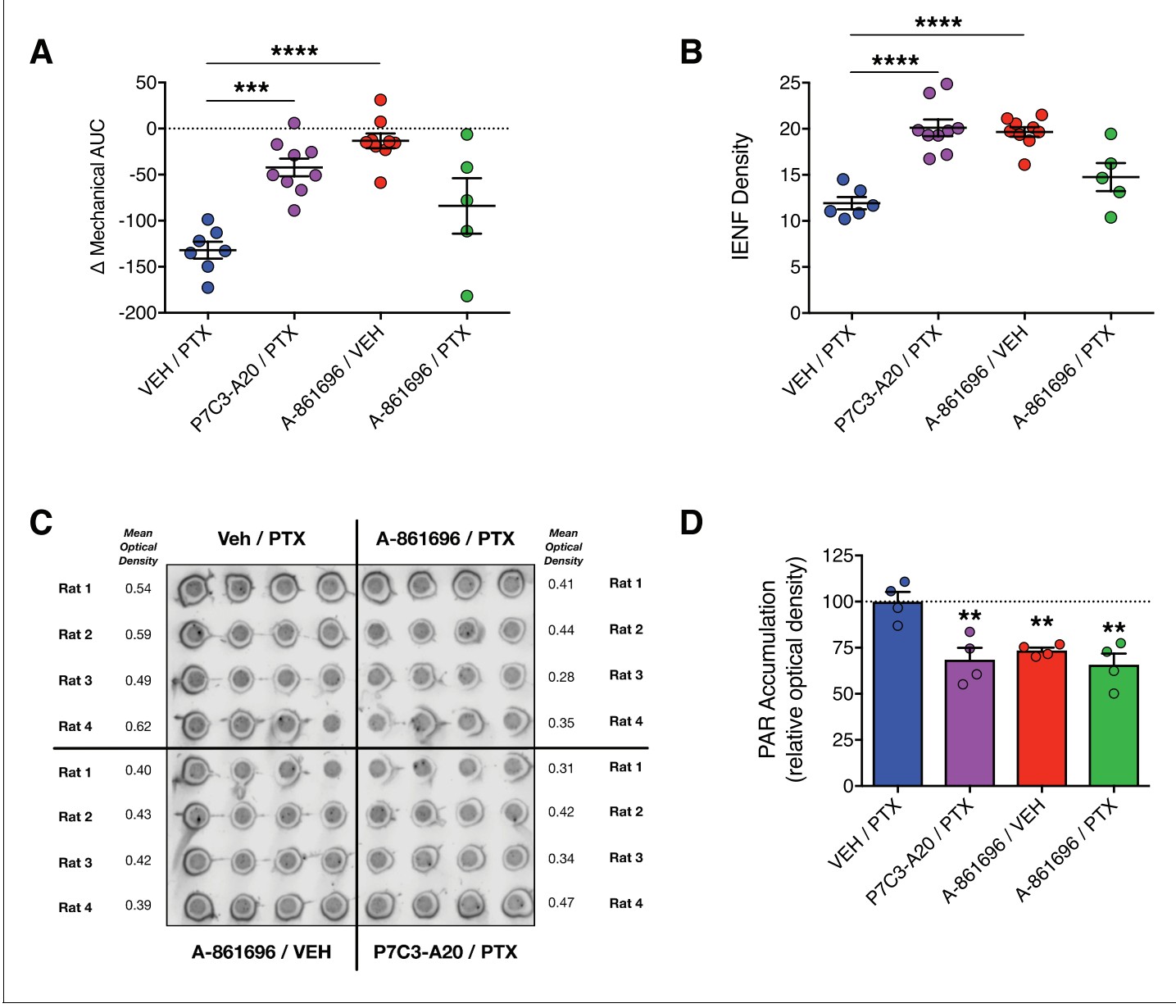

**Figure 4.** Inhibition of PARP does not prevent PTX-induced peripheral neuropathy in rats. (**A**) Mechanical AUCs of rats treated with vehicle, P7C3-A20 (20 mg/kg/day, i.p.), or A-861696 (50 mg/kg/day, i.p.). Horizontal lines represent mean AUC ±SEM calculated from individual rat AUC values shown as small circles. (**B**) IENF densities from hindpaw biopsies collected on day 12. Horizontal lines represent the mean ±SEM calculated from individual rat IENF densities from each group (circles), n = 5–9 rats/group. (**C and D**) Dot-blot analysis of poly(ADP-ribose) (PAR) accumulation in lumbar DRG neurons of rats treated with PTX and either vehicle, P7C3-A20, or A-861696. Individual rat DRG homogenates were run in quadruplicate and the relative optical density was calculated for each rat. Bars represent the mean ±SEM, n = 4 rats/group. ****p<0.0001, ***p<0.001, **p<0.01 vs. Veh/PTX by one-way ANOVA with Dunnett's post-hoc test (**A, B, D**).

DOI: https://doi.org/10.7554/eLife.29626.013

The following figure supplement is available for figure 4:

**Figure supplement 1.** Comparison of the effects of P7C3-A20 and A-861696 (PARP inhibitor) on PTX-induced peripheral neuropathy.
DOI: https://doi.org/10.7554/eLife.29626.014

NAD$^+$ levels in the hindpaw and sciatic nerve, which was abolished by P7C3-A20 treatment (*Figure 7A and B*). NAD$^+$ levels in the DRG were unaffected by either drug (*Figure 7C*). Collectively, these data further support the hypothesis that P7C3-A20 augments NAMPT-mediated NAD production in peripheral sensory neurons in response to the damaging effects of PTX.

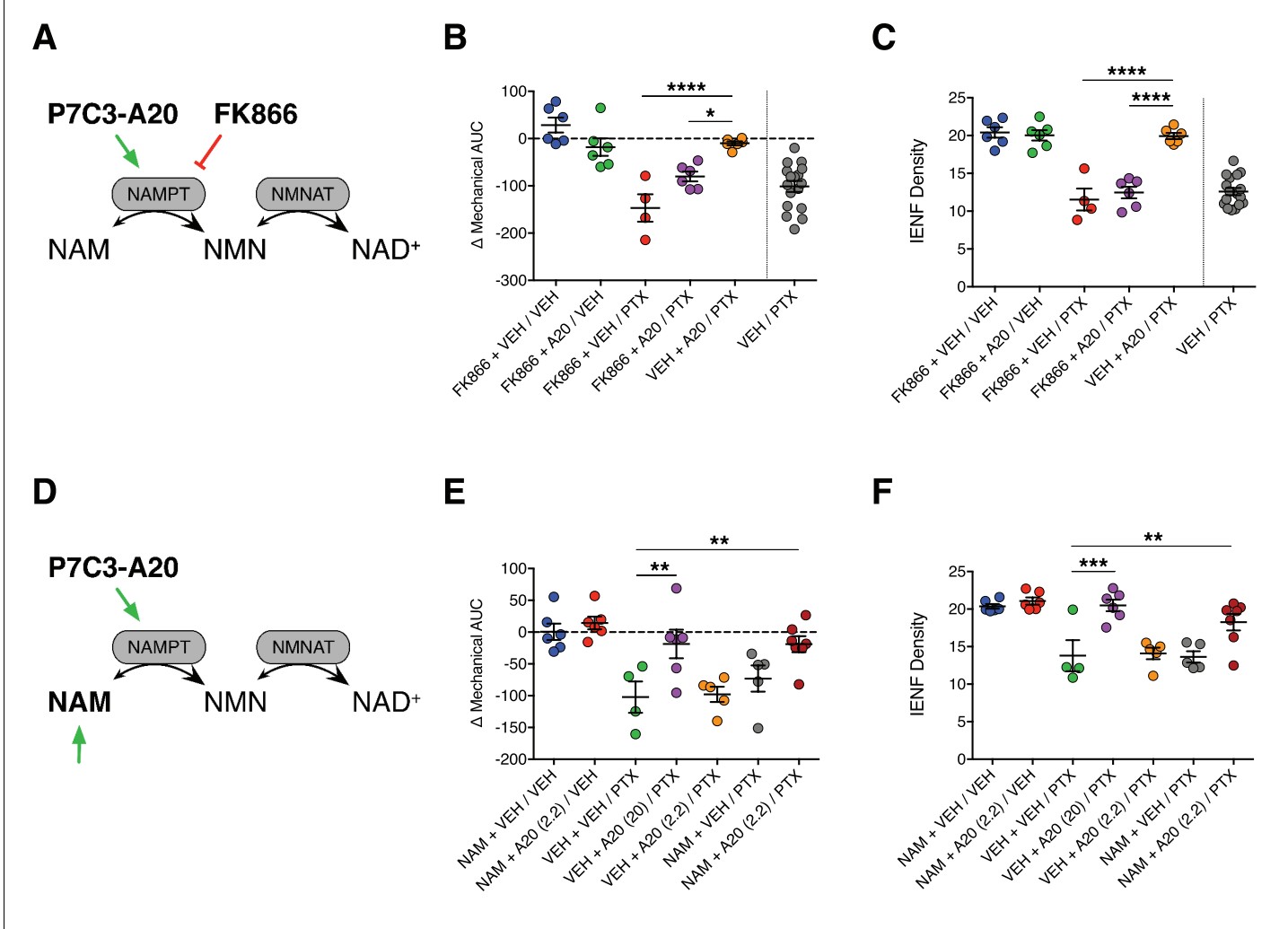

**Figure 5.** Stimulation of NAMPT is required to prevent PTX-induced damage to peripheral nociceptive neurons. (**A**) FK866 (0.5 mg/kg, i.p., b.i.d.) was injected in rats to antagonize NAMPT stimulation by P7C3-A20 (10 mg/kg/day, i.p.) in response to PTX. (**B**) Individual rat mechanical AUCs. (**C**) IENF densities from hindpaw biopsies collected on day 12. Horizontal lines represent the mean ±SEM calculated from individual rat AUCs or IENF densities in each treatment group shown as small circles. ****p<0.0001, *p<0.05 vs. Veh + A20/PTX by one-way ANOVA with Sidak's post-hoc test, n = 4–6 rats/group. (**D**) Exogenous NAM (150 mg/kg/day, s.c.) was administered to potentiate an ineffective dose of P7C3-A20 (2.2 mg/kg/day, i.p.) against PTX. (**E** and **F**). Scatter plots of mechanical AUCs (**E**) and IENF densities (**F**). ***p<0.001, **p<0.01 vs. Veh + Veh/PTX by one-way ANOVA with Sidak's post-hoc test, n = 4–7 rats/group.

DOI: https://doi.org/10.7554/eLife.29626.015

The following figure supplements are available for figure 5:

**Figure supplement 1.** Inhibition of NAMPT with the selective inhibitor, FK866.
DOI: https://doi.org/10.7554/eLife.29626.016

**Figure supplement 2.** Modulation of NAMPT with its substrate, nicotinamide (NAM).
DOI: https://doi.org/10.7554/eLife.29626.017

## P7C3-A20 does not alter cancer cell proliferation or chemosensitivity of cancer cells to PTX

The striking ability of P7C3-A20 to attenuate PTX-induced neuropathy prompted a careful evaluation of its effects on the anti-proliferative and cytotoxic effects of PTX on cancer cells. Thus, we examined the effect of P7C3-A20 on PTX-induced growth inhibition of several cancer cells lines, including: HeLa (cervical), SK-OV-3 (ovarian), MDA-MB-231 (breast), Panc-1 (pancreatic), and SK-N-BE(2) (neuroblastoma) cells. P7C3-A20 did not change the anti-proliferative or cytotoxic potency or efficacy of

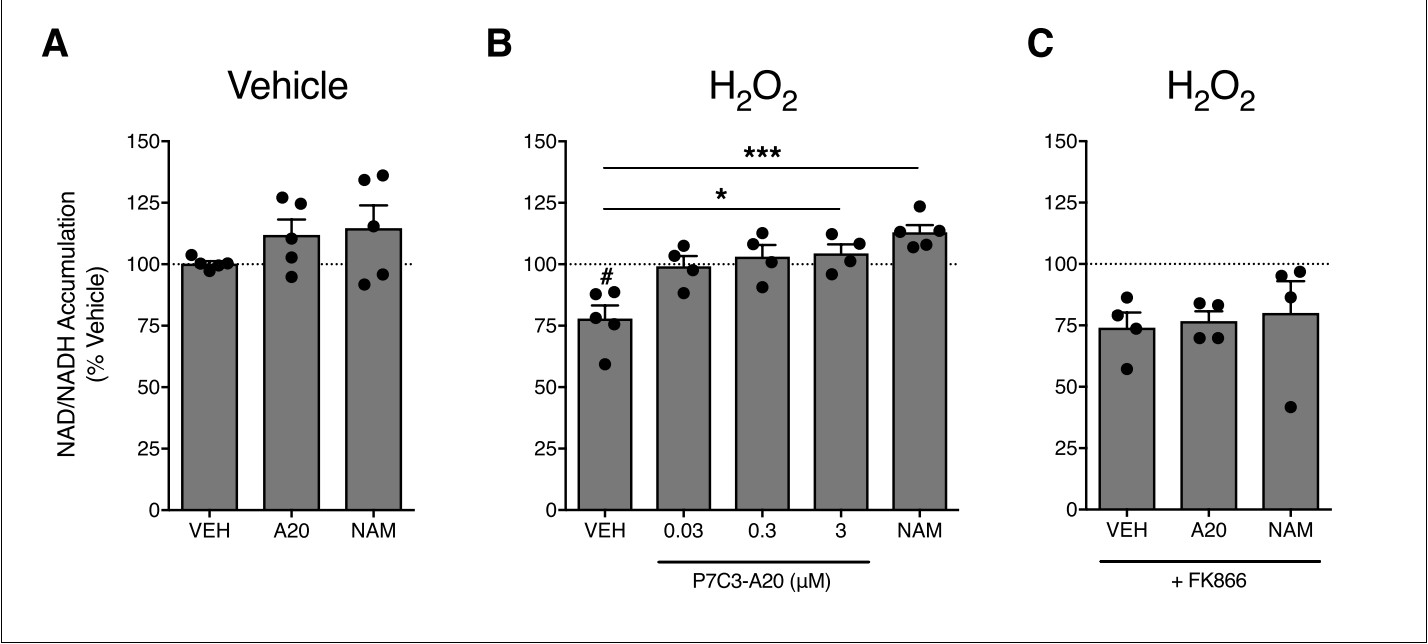

**Figure 6.** P7C3-A20 stimulates NAD recovery in response to depletion with $H_2O_2$ treatment in vitro. (**A**) Effects of vehicle, P7C3-A20 (3 µM), or NAM (1 mM) on intracellular NAD production under basal conditions. A1A1 cells were pre-treated with vehicle (30 min) followed by indicated treatment (60 min). (**B and C**) Effects of vehicle, P7C3-A20, or NAM (1 mM) on intracellular NAD production in response to pretreatment with $H_2O_2$ (200 µM, (**B**). FK866 (10 nM) was co-administered with P7C3-A20 (3 µM) or NAM (1 mM) after $H_2O_2$ pretreatment (**C**). Treatment conditions were performed in quadruplicate for each experiment. Bars represent mean NAD accumulation ±SEM expressed as a percentage of vehicle. Circles are mean NAD values from independent experiments. #$p < 0.05$ vs. Veh/Veh or ***$p < 0.001$, *$p < 0.05$ vs. $H_2O_2$/Veh by one-way ANOVA with Dunnett's post-hoc test, n = 4–5 independent experiments.

DOI: https://doi.org/10.7554/eLife.29626.018

PTX, nor did it alter the microtubule-stabilizing properties of PTX (*Figure 8A and B* and *Figure 8— figure supplement 1A and B*). For most of the cell lines, P7C3-A20 alone did not alter proliferation rate, however, at the highest concentration used, it slightly increased growth of cultured MDA-MB-231 cells (*Figure 8C* and *Figure 8*, *Figure 8—figure supplement 1C*).

We next evaluated the effect of P7C3-A20 on PTX-mediated antitumor activity in vivo using MDA-MB-231 xenografts in female athymic nude mice. We also measured mechanical allodynia in these mice. We elected to implant MDA-MB-231 breast cancer cells on the bases that these tumors are sensitive to PTX treatment in vivo and since P7C3-A20 slightly increased the proliferation rate of these cells in vitro. Tumors were allowed to grow for 4 weeks following implantation. Mice were then treated with daily injections of P7C3-A20 (20 mg/kg, i.p.) or vehicle and PTX (11.7 mg/kg, i.p.) or vehicle on days 0, 2, 4. PTX treatment markedly decreased tumor growth as expected. P7C3-A20 did not alter the antitumor effects of PTX, nor alter tumor growth when administered alone (*Figure 8D*). As with its effect in rats, P7C3-A20 completely prevented PTX-induced mechanical allodynia in the tumor-bearing mice (*Figure 8E* and *Figure 8—figure supplement 2*). Collectively, these results demonstrate that P7C3-A20 treatment can attenuate the deleterious effects of PTX on peripheral nociceptive neurons and mitigate peripheral neuropathic pain without compromising the desired antitumor effects.

## Discussion

Here, we discovered that the aminopropyl carbazole, P7C3-A20, displayed remarkable neuroprotective efficacy in an aggressive CIPN model of PTX-induced peripheral neuropathy. High-dose PTX treatment (cumulative dose of 35 mg/kg, i.p.) produced transient but substantial average weight loss of 16% and average leukocyte depletion of 60%. Indices of general health were poor and attrition due to PTX toxicity (death or weight loss >20%) was 25%. As is typical of CIPN (*Cata et al.,*

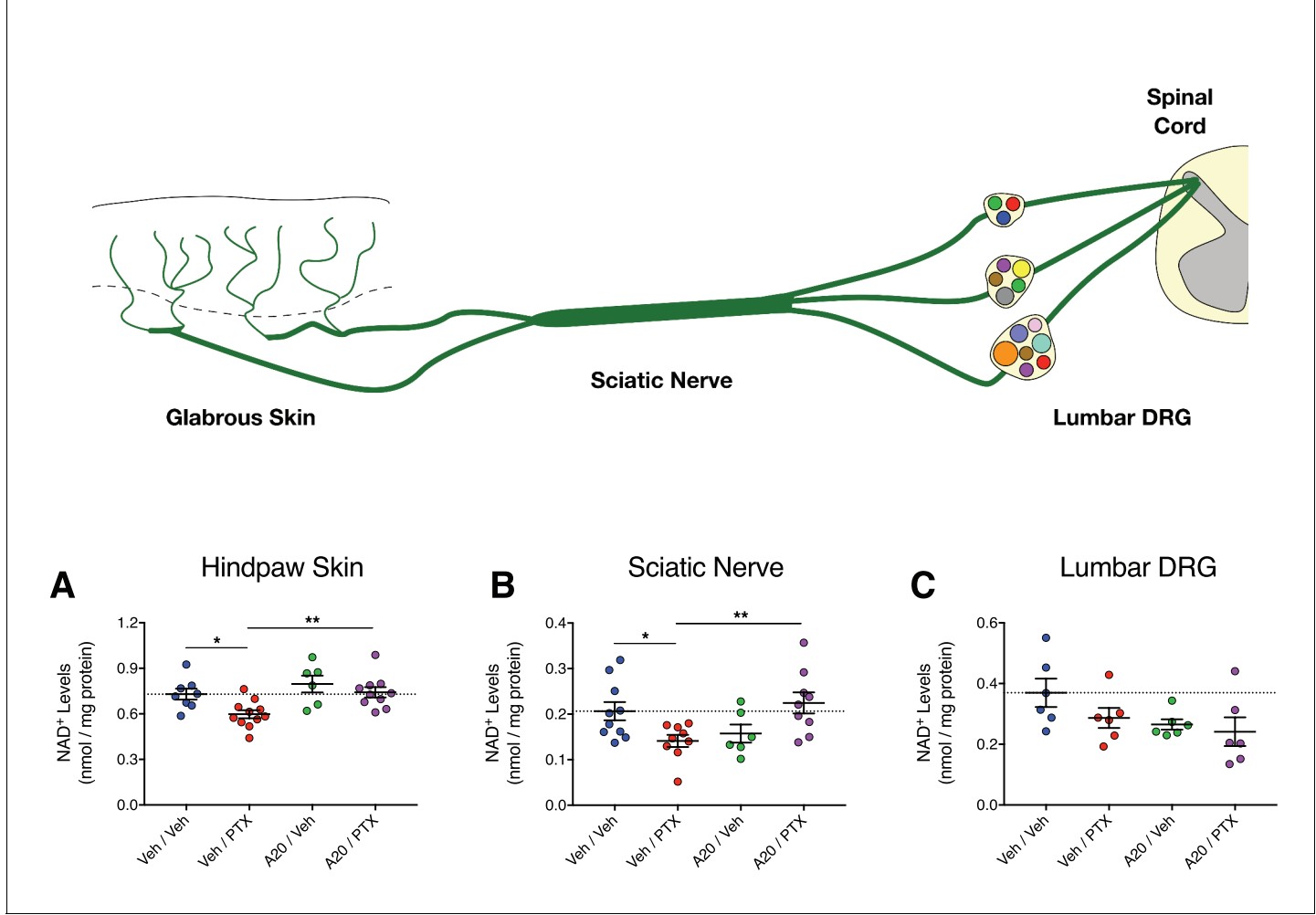

**Figure 7.** P7C3-A20 stimulates NAD recovery in peripheral neurons damaged by PTX in vivo. (A–C) Tissue NAD$^+$ levels in glabrous hindpaw skin (A), sciatic nerve (B), and lumbar DRG (C) collected from treated rats on day 10 of the experimental paradigm. As before, rats were treated daily with P7C3-A20 (20 mg/kg/day; i.p.) or vehicle beginning on day −2 through day 9, with PTX (11.7 mg/kg, i.p.) or vehicle injected on days 0, 2, and 4. On day 10, tissue samples were collected for homogenization and subsequent metabolite analysis using the NAD/NADH-Glo Assay (Promega, Madison, WI). Tissue NAD$^+$ levels were normalized to total protein content. Bars identify the mean metabolite levels (nmol/mg protein)±SEM for each treatment group. Circles represent individual rat tissue NAD$^+$ levels. **$p<0.01$, *$p<0.05$ vs. specified group by one-way ANOVA with Sidak's post-hoc test, n = 6–11 tissue samples/group.

DOI: https://doi.org/10.7554/eLife.29626.019

*2006*; *Dougherty et al., 2004*; *Nahman-Averbuch et al., 2011*), PTX treatment also produced marked peripheral neuropathy evidenced by prolonged changes in the sensitivity to mechanical, cold and heat stimuli, by prolonged degeneration of IENFs, and by increased expression of the neuronal injury marker, ATF3, in the DRG. Daily treatment with P7C3-A20 (≥10 mg/kg/day, i.p.), beginning a few days before PTX administration, greatly reduced or eliminated the behavioral and anatomical signs of peripheral neuropathy. The neuroprotective effects were replicated consistently over several independent, blinded behavioral experiments and across two different species. P7C3-A20 also demonstrated superior efficacy in our aggressive model as compared with PARP inhibitor, A-861696, which previously was shown to protect against cisplatin-, oxaliplatin-, and vincristine-induced peripheral neuropathy (*Brederson et al., 2012*; *Ta et al., 2013*). P7C3-A20 also improved the overall health of the PTX-treated rats and reduced attrition to zero. Importantly, P7C3-A20 treatment did not reduce the antiproliferative or cytotoxic efficacy of PTX in cell culture models nor its antitumoral activity in an aggressive xenograft model in vivo.

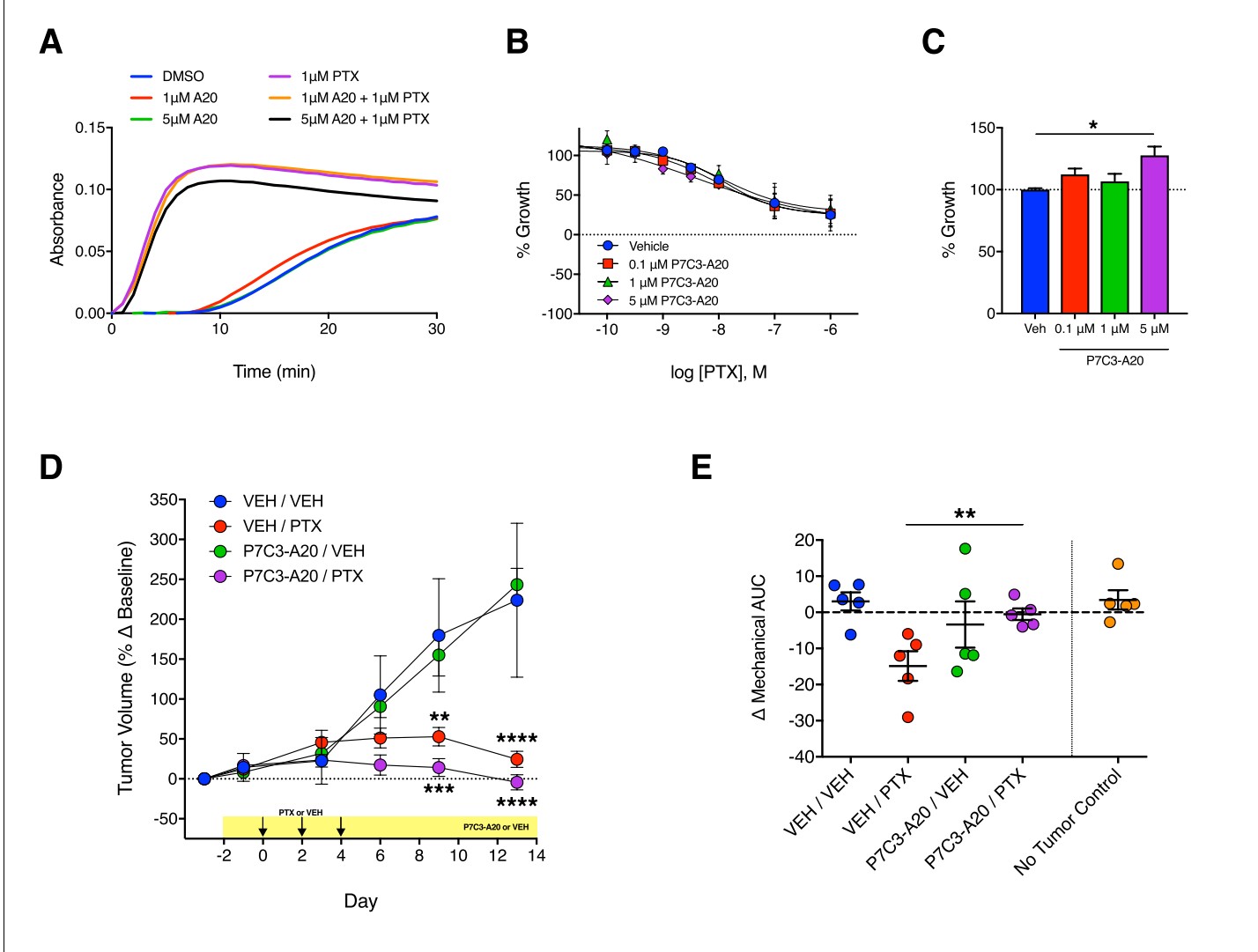

**Figure 8.** Antitumoral efficacy of PTX is maintained in the presence of P7C3-A20. (**A**) Tubulin polymerization curves corresponding to addition of P7C3-A20 with and without PTX (n = 3 independent experiments). (**B**) Concentration-dependent anti-proliferation of PTX (48 hr) in MDA-MB-231 breast cancer cells pretreated (1 hr) with vehicle or P7C3-A20 (0.1–5 µM). (**C**) Effects of P7C3-A20 treatment only on growth of MDA-MB-231 cells. *p<0.05 vs. Veh by one-way ANOVA followed by Dunnett's post-hoc test, n = 3 independent experiments. (**D**) Timecourse of changes in MDA-MB-231 tumor volumes in female athymic nude mice treated with P7C3-A20 (20 mg/kg/day, i.p.) or vehicle and PTX (11.7 mg/kg, i.p.) as indicated. ****p<0.0001, ***p<0.001, **p<0.01 vs. Veh/Veh by two-way mixed-effect ANOVA with Dunnett's post-hoc test, n = 8–9 tumors/group. (**E**) Mechanical AUCs from tumored mice. Control mice lacking tumors were tested concurrently with the tumored mice. **p<0.01 by one-way ANOVA followed by Sidak's post-hoc test, n = 5 mice/group.

DOI: https://doi.org/10.7554/eLife.29626.020

The following figure supplements are available for figure 8:

**Figure supplement 1.** P7C3-A20 does not alter the anti-proliferative or microtubule-stabilizing capacity of PTX in vitro.

DOI: https://doi.org/10.7554/eLife.29626.021

**Figure supplement 2.** Effect of P7C3-A20 on PTX-induced mechanical allodynia and body weight in mice with implanted MDA-MB-231 tumor xenografts.

DOI: https://doi.org/10.7554/eLife.29626.022

P7C3-A20 is a first-in-class stimulator of NAMPT (*Wang et al., 2014*), the rate-limiting enzyme in the salvage pathway for NAD production (*Belenky et al., 2007*). Our data suggest that NAMPT is critical for the neuroprotective efficacy of P7C3-A20 against PTX-induced sensory neuron damage and peripheral neuropathic pain. The effects of P7C3-A20 to prevent PTX-induced allodynia and IENF degeneration were completely blocked by FK866, a well-characterized and selective inhibitor

of NAMPT (*Hasmann and Schemainda, 2003*). Supplementation with the NAMPT substrate, NAM, augmented the efficacy of an ineffective dose of P7C3-A20 to that equivalent to a maximal dose. We found that P7C3-A20 increased NAD production, in an FK866-dependent manner, in neuronal cells depleted of NAD by treatment with $H_2O_2$. Furthermore, we determined that P7C3-A20 treatment prevented PTX-induced deficits in peripheral neuron NAD levels in vivo, results consistent with recent work showing that P7C3-A20 stimulated NAD levels in brain tissue following ischemia-induced deficit (*Loris et al., 2017*; *Wang et al., 2016a*). Taken together, these results are consistent with NAMPT as a target for P7C3-A20.

There is increasing evidence that stimulation or maintenance of NAD biosynthesis is neuroprotective in a variety of experimental models of neurodegeneration (*Cantó et al., 2015*; *Verdin, 2015*). For example, due to increased cytosolic levels of chimeric nicotinamide mononucleotide adenylyl-transferase1, a critical enzyme for NAD biosynthesis, Wld$^S$ mice exhibit marked protection from injury-induced axonal degeneration (*Araki et al., 2004*; *Wang et al., 2005*) and are also resistant to PTX-induced peripheral neuropathy (*Wang et al., 2002*). Additionally, administration of the NAD precursor, nicotinamide riboside (NR), reduces the sensory neuropathy that occurs in diabetic mice (*Trammell et al., 2016*) and in female rats treated with PTX (*Hamity et al., 2017*). Moreover, P7C3 compounds have strong neuroprotective efficacy in several models of neurodegeneration, including Parkinson's disease, amyotrophic lateral sclerosis, traumatic brain injury, optic nerve injury, ischemic stroke, and sciatic nerve crush (*De Jesús-Cortés et al., 2012*; *Kemp et al., 2015*; *Loris et al., 2017*; *Oku et al., 2017*; *Tesla et al., 2012*; *Wang et al., 2016a*; *Yin et al., 2014*).

Neurons are energetically demanding cells that rely heavily on oxidative metabolism to meet bioenergetic demand (*Berndt and Holzhütter, 2013*; *Van Laar and Berman, 2013*). IENFs in particular have high-energy requirements as they must undergo continuous remodeling to accommodate the constantly changing epidermis (*Palay, 1956*) and are especially sensitive to reductions in bioenergy production (*Bennett et al., 2014*; *Bennett et al., 2011*). PTX, among many other chemotherapeutic agents, causes substantial mitotoxicity, including deficits in complex I- and II-mediated respiration (*Zheng et al., 2011*; *Zheng et al., 2012*) and diminished capacity to synthesize ATP (*Duggett et al., 2017*; *Janes et al., 2013*). These effects exhaust energy stores that are necessary to sustain neuronal repair mechanisms and maintain viability (*Janes et al., 2013*; *Zhou et al., 2016*) and coincide with spontaneous discharge of nociceptive and non-nociceptive afferent fibers and progressive degeneration of IENFs that collectively contribute to peripheral neuropathy (*Bennett et al., 2011*; *Kleggetveit et al., 2012*; *Xiao and Bennett, 2008*; *Xiao et al., 2011*). In our experiments, we observed persistent IENF degeneration in glabrous hindpaw skin biopsies in response to PTX that was completely reversed by treatment with P7C3-A20. Individual animal IENF density correlated very strongly with their respective behavioral responses. We also determined that P7C3-A20 reversed the reduction in NAD$^+$ levels in glabrous skin by PTX. These results are particularly intriguing in consideration of the high density of mitochondria within IENFs (*Palay, 1956*; *Ribeiro-da-Silva et al., 1991*). However, we cannot rule out the possibility of an indirect effect of P7C3-A20 on IENFs mediated through other cell types in the skin, especially those known to support neurons (e.g. Schwann cells, keratinocytes). Regardless, NAD drives mitochondrial bioenergy production, and thus its availability is absolutely critical to sustain the energy needs of IENFs.

It is notable that P7C3-A20 reduced, but did not completely prevent, PTX-induced expression of the neuronal injury marker, ATF3. This suggests that P7C3-A20 may not prevent damage to peripheral sensory neurons by PTX but instead, by augmenting NAMPT activity, may maintain metabolic output of damaged neurons to allow time for repair mechanisms to reverse PTX-induced toxic damage and reduce the peripheral neuropathy. We did not observe any change in NAD levels in lumbar DRG assessed at day 10 (6 days after the final injection of PTX), but this does not rule out that effects may occur at time points closer to PTX administration.

P7C3-A20 did not alter nociceptive thresholds, general health, or IENF density in rats not treated with PTX and it is noteworthy that we did not observe increases in NAD production by P7C3-A20 in normal neuronal cells. Only when NAD was first depleted by treatment with $H_2O_2$ or PTX did P7C3-A20 stimulate NAMPT-mediated NAD production back to baseline levels. Similarly, previous reports suggest that robust increases in NAD production in response to P7C3-A20 occurs when NAD is depleted in damaged neurons (*Loris et al., 2017*; *Wang et al., 2016a*). These results suggest that P7C3-A20 may only be effective in energy-deficient cells with NAD depletion. Relatively little is known of the mechanisms by which NAMPT is regulated, however, NAMPT activity appears to be

tightly regulated (*Burgos et al., 2009*; *Dölle et al., 2013*; *Takahashi et al., 2010*; *Yoon et al., 2015*) and is known to be subject to negative feedback regulation by NAD (*Burgos and Schramm, 2008*). Efficacy of P7C3-A20 to stimulate NAD production only in NAD-depleted cells suggests that P7C3-A20 acts as an allosteric regulator to augment NAMPT activity such that NAMPT retains negative feedback sensitivity which would avoid accumulation of excessive NAD levels. The ability of P7C3-A20 to stimulate NAD production when levels are low, while limiting overproduction, would likely limit adverse effects associated with excess NAD such as flushing, itching, erythema, nausea, and hepatotoxicity (*Knip et al., 2000*). Although drugs such as NR, which bypass the regulatory control provided by the rate-limiting function of NAMPT in the NAD salvage pathway (*Bieganowski and Brenner, 2004*), may provide similar benefit for treatment of various neurodegenerative conditions, these drugs would be expected to cause overproduction of NAD in most cells, not just those that are energy-compromised.

Although a plethora of agents have been tested clinically, there still is no effective treatment or prevention of CIPN (*Hershman et al., 2014*). One possible reason for the failed translation of therapeutics for CIPN is the selection of anticancer drug dose selected for use in the preclinical animal models. Many preclinical models of PTX-induced peripheral neuropathy utilize a low cumulative dose, ranging from 4 to 8 mg/kg (*Authier et al., 2009*), which converts roughly to 28–56 mg/m$^2$ equivalent human dose (*Nair and Jacob, 2016*). These low doses do not produce leukopenia, and thus do not adequately recapitulate clinical symptomatology associated with CIPN. Clinical doses of PTX that exceed 200 mg/m$^2$ (28.6 mg/kg rat equivalent dose) per single infusion or 1400 mg/m$^2$ (200 mg/kg rat equivalent dose) cumulatively cause dose-limiting neuropathic pain that interferes with daily activities and reduces quality of life (*Lee and Swain, 2006*; *Miltenburg and Boogerd, 2014*). Here, we elected to administer an aggressive, but clinically relevant, cumulative dose of 35 mg/kg PTX to the rats. With this dose, we observed significant weight loss and leukopenia that resulted in an attrition rate of 25%. Rats treated with this PTX regimen exhibited differentially altered nociceptive sensitivities to mechanical, cold, and heat stimulation, which is consistent with observations made in patients receiving PTX (*Cata et al., 2006*; *Dougherty et al., 2004*; *Nahman-Averbuch et al., 2011*). The development of mechanical and cold allodynia as well as heat hypoalgesia occurred concomitantly with significant IENF degeneration observed in hindpaw biopsies. Clinically, IENF loss corresponds with these same sensory deficits in patients with CIPN as well as in patients with other types of peripheral neuropathy (*Oaklander, 2001*; *Petersen et al., 2010*; *Tai et al., 2004*). Peripheral nerve damage also was apparent in lumbar DRG perikarya as evidenced by up-regulation of ATF3, consistent with earlier studies using higher-dose PTX models (*Liu et al., 2010*; *Peters et al., 2007a*; *Verheyen et al., 2012*). Collectively, the high cumulative dose of PTX that we utilized clearly produced a severe peripheral neuropathy-like phenotype in the rats that is consistent with the clinical presentation of CIPN. In consideration of the similarities to patient symptoms and histopathology, this work supports the use of higher-dose models for future preclinical evaluation of potential CIPN interventions.

The PARP inhibitor, A-861696, was ineffective in our model of PTX-induced peripheral neuropathy. In the previous studies where A-861696 displayed some efficacy to reduce peripheral neuropathy, rodents were treated with cumulative doses of vincristine, cisplatin, or oxaliplatin that did not result in significant weight loss or leukopenia and that produced a modest mechanical allodynia (*Brederson et al., 2012*; *Ta et al., 2013*). As discussed above, the ineffectiveness of A-861696 here may due to the severity of peripheral neuropathy produced by the aggressive dose of PTX used in our model. However, a recent paper also reported a lack of efficacy using veliparib, the enantiomer of A-861696, in models of optic nerve injury and spinal cord injury (*Wang et al., 2016b*). This could be a consequence of compartmentalized NAD recovery by A-861696. PARPs predominantly localize in the nucleus and rely on activities of NAMPT and NMNAT1 to produce NAD that can be used to generate PAR polymers that coordinate DNA repair (*Schreiber et al., 2006*). Peripheral sensory neurons can extend over 1 m in length, in which case the distal terminals are far removed from the effects of changes in nuclear PARP inhibition and reduced NAD consumption. NAMPT, however, localizes in the cytosol and potentially in mitochondria as well as in the nucleus (*Belenky et al., 2007*). Thus, the efficacy observed by stimulation of NAMPT with P7C3-A20 may be such that it can elevate NAD concentrations neuron-wide, including in the distal terminals of peripheral sensory neurons where energy repletion may improve IENF vitality.

An unexpected observation was the clear improvement in the general health of the animals receiving P7C3-A20 with PTX. The indices used to qualitatively assess the animals' well-being included markers of deficits in sensory and motor function, pain-depressed behaviors, autonomic deficits, and gastrointestinal function. Although not explicitly tested here, it is possible that increased NAD could contribute to the health improvements produced by P7C3-A20. NAMPT-mediated NAD recovery maintains NAD-dependent enzymes that regulate, for example, cellular metabolism or the adaptive responses to oxidative stress that preserve cell vitality and function (*Garten et al., 2015*). Additionally, increased NAD levels have been shown to facilitate improvements in several tissue types, such as an enhanced oxidative capacity in skeletal muscle of obese mice (*Cantó et al., 2012*), elevated insulin sensitivity in livers of diabetic mice (*Yoshino et al., 2011*), and improve mineral density in bones of aged mice (*Mills et al., 2016*). Therefore, many of the observed improvements from PTX-induced toxicity could be a direct result of enhanced NAMPT activity.

Increases in cellular NAD levels, produced by overexpression of NAMPT or exogenous application of NAD metabolites, have been associated with enhanced cancer growth and greater resistance to anticancer agents (*Cairns et al., 2011*; *Chiarugi et al., 2012*) and inhibition of NAMPT is currently under evaluation as an adjuvant in chemotherapy regimens. If NAMPT is to be a viable target for drugs to prevent CIPN, it is critical to determine whether stimulation of NAMPT with P7C3-A20 enhances tumor growth or whether it interferes with the anticancer activity of PTX. Importantly, we found that P7C3-A20 treatment did not alter the antitumor efficacy of PTX treatment in mice with transplanted xenografts of triple-negative breast cancer (MDA-MB-231 tumors). It is noteworthy that in these tumored mice, P7C3-A20 also completely prevented the mechanical allodynia that occurred in response to PTX. We also examined the effect of P7C3-A20 on growth of 5 distinct cancer cells lines. P7C3-A20 did not alter in vitro cancer growth, with the exception of a slight increase in the growth of MDA-MB-231 cells, which were unaffected by P7C3-A20 treatment in vivo. The overall lack of an appreciable effect on cancer cell growth may be related to the ability of P7C3-A20 to replenish cellular NAD to normal levels, but not to overproduce NAD.

The effectiveness of P7C3-A20 in our aggressive model of PTX-induced peripheral neuropathy highlights NAMPT-mediated NAD salvage as a new therapeutic target for CIPN. This is of high clinical importance, as cancer patients with debilitating peripheral neuropathy induced by PTX or other anticancer agents must endure diminished quality of life due to the lack of an effective treatment for CIPN. This study also contributes to the increasing evidence that maintenance of cellular NAD supply protects against neuronal damage and neurodegeneration (*Cantó et al., 2015*; *Gerdts et al., 2016*; *Verdin, 2015*). Accordingly, clinical investigation may be warranted for the first-in-class NAMPT stimulators like P7C3-A20 as an intervention for CIPN.

## Materials and methods

**Key resources table.**

| Reagent type (species) or resource | Designation | Source or reference | Identifiers | Additional information |
|---|---|---|---|---|
| cell line (human) | HeLa | ATCC | ATCC Cat# CRM-CCL-2; RRID:CVCL_0030 | Cell line maintained in A. Risinger lab. Authenticated by S TR profiling (Genetica). |
| cell line (human) | Panc-1 | ATCC | ATCC Cat# CRL-1469; RRID:CVCL_0480 | Cell line maintained in A. Risinger lab. Authenticated by STR profiling (Genetica). |
| cell line (human) | MDA-MB-231 | ATCC | ATCC Cat# CRM-HTB-26; RRID:CVCL_0062 | Cell line maintained in A. Risinger lab. Authenticated by STR profiling (Genetica). |
| cell line (human) | SK-OV-3 | ATCC | ATCC Cat# HTB-77; RRID:CVCL_0532 | Cell line maintained in A. Risinger lab. Authenticated by STR profiling (Genetica). |

*Continued on next page*

*Continued*

| Reagent type (species) or resource | Designation | Source or reference | Identifiers | Additional information |
|---|---|---|---|---|
| cell line (human) | SK-N-BE(2) | ATCC | ATCC Cat# CRL-2271; RRID:CVCL_0528 | Cell line maintained in A. Risinger lab. Authenticated by STR profiling (Genetica). |
| cell line (rat) | A1A1 | PMID:8190100 | | Cell line maintained in W. Clarke lab |
| antibody | anti-PGP9.5 (mouse monoclonal) | Encor Biotechnology | EnCor Biotechnology Cat# MCA-BH7-AP; RRID:AB_2572395 | (1:1000) |
| antibody | anti-collagen type IV (goat polyclonal) | Southern Biotechnology | Southern Biotechnology Cat# 1340–01 | (1:200) |
| antibody | anti-ATF3 (rabbit polyclonal) | Santa Cruz Biotechnology | Santa Cruz Biotechnology Cat# sc-188; RRID:AB_2258513 | (1:400) |
| antibody | anti-NeuN (mouse monoclonal) | Millipore (Chemicon) | Millipore Cat# MAB377; RRID:AB_2298772 | (1:250) |
| antibody | anti-poly(ADP-ribose) 10 hr (mouse monoclonal) | Enzo Life Sciences | Enzo Life Sciences Cat# ALX-804–220, RRID:AB_2272987 | (1:300) |
| antibody | anti-β-tubulin (mouse monoclonal) | Sigma | Sigma Cat# T8328; RRID:AB_1844090 | (1:400) |
| antibody | anti-mouse Alexa Fluor 488 secondary (donkey polyclonal) | Jackson ImmunoResearch | Jackson ImmunoResearch Code# 715-545-150 | (1:500) |
| antibody | anti-rabbit Alexa Fluor 594 secondary (donkey polyclonal) | Jackson ImmunoResearch | Jackson ImmunoResearch Code# 711-585-152 | (1:500) |
| antibody | anti-goat Alexa Fluor 594 secondary (donkey polyclonal) | Jackson ImmunoResearch | Jackson ImmunoResearch Code# 705-585-147 | (1:500) |
| antibody | anti-mouse IRDye 800CW secondary (goat polyclonal) | LI-COR Biosciences | LI-COR Biosciences Cat# 827–08364; RRID:AB_10793856 | (1:10,000) |
| antibody | anti-mouse FITC secondary (sheep polyclonal) | Sigma | Sigma Cat# F6257; RRID:AB_259692 | (1:200) |
| other | DAPI stain | Sigma (Roche) | Sigma Cat# 10236276001 | (0.1 µg/ml) |
| other | VECTASHIELD HardSet Mounting Medium | Vector Laboratories | Vector Laboratories Cat# H-1400; RRID:AB_2336787 | |
| commercial assay, kit | NAD/NADH-Glo Assay | Promega Corporation | Promega Cat# G9071 | |
| commercial assay, kit | Tubulin polymerization assay, OD based - Porcine | Cytoskeleton, Inc. | Cytoskeleton Cat# BK006P | |

## Animals

Adult male Sprague-Dawley rats (RRID:RGD_734476) weighing 250–300 g, and female athymic nude *Foxn1^{nu}* mice (RRID:MGI:2680668) weighing 20–25 g, were used in this study. Rats were housed in

groups of 3 and mice were housed in groups of 5. Animals were maintained on a 12 hr light-dark cycle with ambient temperatures between 20°C and 22°C. Food and water were available ad libitum. Animals were labeled numerically with permanent marker on the tail to monitor each animal throughout each study, as well as to allow drug dose/volume administration based on body weight. Body weight was recorded daily. This study was performed in strict accordance with the recommendations in the Guide for the Care and Use of Laboratory Animals of the National Institutes of Health. The animal study protocol (#20130051AR) was approved by the Institutional Animal Care and Use Committee of the University of Texas Health Science Center at San Antonio and conformed to International Association for the Study of Pain (IASP) and federal guidelines.

## Drug preparation and administration

To conduct and maintain blinded experiments, drugs were prepared fresh daily in coded vials (2 ml/kg) by an individual not involved in administering injections or behavioral testing. Every day, a spreadsheet was prepared indicating the coded vial and injections volume based upon body weight for each animal. Two different experimenters prepared syringes using the coded spreadsheet and then administered injections to animals. Following injections, animals were returned to their storage room and checked for signs of distress before leaving them for the night.

Paclitaxel (PTX; LC Laboratories, Woburn, MA) first was dissolved completely in dehydrated ethanol. An equal volume of Kolliphor (1:1) then was added to make a 12 mg/ml stock solution. The PTX solution was then diluted with sterile PBS (1:3). Either PTX or vehicle, at an equivalent volume (EtOH/Kolliphor/PBS, 1:1:6), was injected intraperiotoneally (i.p.) in rats and mice at a dose of 11.7 mg/kg on three alternate days (days 0, 2, and 4). Thus, the final cumulative dose of PTX was 35 mg/kg per animal. Animals were weighed every day and monitored daily for signs of distress. Any animal that was in distress or had substantial weight loss (>20%) was removed from the study and euthanized. On behavioral testing days, blinded experimenters recorded observations and behaviors associated with the general health for each animal (*Authier et al., 2000*; *Cavaletti et al., 1995*; *Wozniak et al., 2011*).

For initial experiments, P7C3-A20 and P7C3-S321 were graciously provided by Andrew Pieper (Univ. Iowa). For subsequent experiments, P7C3-A20 was provided by AbbVie, Inc. (North Chicago, IL) P7C3-A20 (or P7C3-S321) was dissolved in 1 vol of DMSO. Four volumes of Kolliphor were added and the solution was vortexed vigorously. The solution was then diluted with 10 volumes of sterile PBS, vortexed, then placed into a 37°C dry bath prior to syringe preparation. The final working stocks were as follows: 10 mg/ml (20 mg/kg dose), 5 mg/ml (10 mg/kg dose), 3.3 mg/ml (6.6 mg/kg dose), and 1.1 mg/ml (2.2 mg/kg dose). Control animals received an equivalent volume of vehicle (DMSO/Kolliphor/PBS, 1:4:10).

A-861696 (provided by AbbVie, Inc.) was dissolved directly in PBS at a concentration of 35 mg/ml. Rats were injected daily at a dose of 50 mg/kg until sacrifice. FK866 (AdooQ Bioscience, Irvine, CA) was dissolved in DMSO (25 mg/ml) and stored at −20°C until ready for use. After thawing, FK866 was diluted with 4 volumes of Kolliphor followed by 10 volumes of PBS (same vehicle as P7C3-A20). Final stock solution was 1.67 mg/ml. Every day until sacrifice, rats were injected twice per day with either FK866 (1.0 mg/kg/day, i.p., b.i.d.) or vehicle, with the first injection given at the same time as P7C3-A20 and the second injection given 6 hr later (*Khan et al., 2006*; *Song et al., 2014*). Nicotinamide (NAM; Sigma, St. Louis, MO) was dissolved directly in PBS at a concentration of 50 mg/ml. Rats were injected subcutaneously (s.c.) with NAM at a dose of 150 mg/kg daily until sacrifice.

## Leukocyte counts

Whole blood samples (50 μl) were obtained from the ventral saphenous vein and transferred into EDTA-coated Microvette CB 300 collection tubes (Kent Scientific, Torrington, CT). To minimize trauma, blood collection alternated between the left and right ventral saphenous veins. After collection, whole blood was diluted 1:20 in Turk's solution (Gentian violet in 2% glacial acetic acid) for erythrocyte lysis. Leukocytes were counted with a hemocytometer and the final cell count was determined with the following equation: *Leukocytes/μl blood = (d x C) / (g x v)*, where $d$ is the dilution factor, $C$ is the total number of cells counted, $g$ is the number of grids, and $v$ is the volume factor.

## Determination of P7C3-A20 and P7C3-S321 levels in plasma

Trunk blood was collected immediately upon decapitation (16 hr after the final injection of P7C3-A20 or vehicle) into a $K_2$EDTA Vacutainer blood collection tube (Becton-Dickinson, Franklin Lakes, NJ) and placed on ice. Samples were then centrifuged at 3000 x g for 5 min to separate the plasma from leukocytes and erythrocytes. Plasma was transferred to cryovials, frozen, and shipped on dry ice to Noelle Williams in the Department of Biochemistry at the University of Texas Southwestern Medical Center for LC-MS/MS analysis.

## Mechanical stimulation assay

To assess the sensitivity of an animal's hindpaw to noxious mechanical stimulation, paw withdrawal threshold (PWT) was evaluated using an electronic Von Frey aesthesiometer equipped with 0.8 mm rigid Supertip filaments (IITC Life Science, Inc., Woodland Hills, CA). Animals were placed in clear plastic observation boxes atop a metal mesh floor. After 30 min acclimation, the aesthesiometer with attached filament was positioned to stimulate the mid-plantar region of the rodent hindpaw, and the force (in grams) required to elicit a paw withdrawal response was displayed on a digital screen and recorded. The mean baseline PWT for naive rats and mice was $44.20 \pm 1.29$ g and $4.43 \pm 0.46$ g, respectively. PWT measurements were taken at least 30 s apart. At least six measurements were recorded per animal per testing day and the mean value was used for statistical analysis. As indicated above, experimenters were blinded to the treatment allocation. Additionally, animal testing order was randomized for each testing day.

## Cold stimulation assay

To assess the sensitivity of an animal's hindpaw to noxious cold stimulation, paw withdrawal latency (PWL) to application of a cold stimulus to the plantar surface of the hindpaw was measured according to a protocol adapted from that described previously (Brenner et al., 2012). Briefly, rats were placed in plastic observation boxes atop 1/8' tempered glass flooring. After 30 min acclimation, a 20-ml syringe (needle end cut off) tightly packed with crushed dry ice was pressed firmly against the glass floor directly beneath the plantar surface of the hindpaw. The latency (in seconds) for the cold stimulus to elicit a paw withdrawal response was timed with a stopwatch and recorded. The mean baseline PWL for naïve rats was $12.53 \pm 0.33$ s. A cutoff of 25 s was used to prevent tissue injury. PWL measurements were taken at least 2 min apart. At least four measurements were recorded per animal per testing day and the mean value was used for statistical analysis. As indicated above, experimenters were blinded to the treatment allocation. Additionally, animal testing order was randomized for each testing day.

## Heat stimulation assay

Paw withdrawal responses to a heat stimulus were measured according to a protocol adapted from that described previously (Hargreaves et al., 1988). Briefly, rats were placed in plastic observation boxes atop temperature-controlled (30°C) glass flooring. After 30 min acclimation, the ventral mid-plantar surface of the rat hindpaw was exposed to a radiant heat source from a thermal stimulator (RRID:SCR_012152) through the glass floor, causing a steady increase in the temperature of the hindpaw. PWL was automatically determined with a photoelectric cell and recorded. The intensity of the heat source was adjusted at the start (day 0) of the experiment such that mean baseline PWL were $10.75 \pm 0.40$ s; cutoff was set for 20 s. PWL measurements were taken at least 60 s apart. At least four measurements were recorded per animal per testing day and the mean was considered for statistical analysis. As indicated above, experimenters were blinded to the treatment allocation. Additionally, the animal testing order was randomized for each testing day.

## Immunohistochemistry

L4-L6 dorsal root ganglia (DRG) and 5 mm paw biopsies from rats were dissected at sacrifice, washed in ice-cold PBS, and immersion-fixed in 4% paraformaldehyde in 0.1M phosphate buffer (PB) at 4°C for 1 hr. Tissue was washed $3 \times 15$ min in PB, cryopreserved in 10% sucrose at 4°C overnight, in 30% sucrose at 4°C overnight, then stored at $-20$°C until sectioned. To prepare for sectioning, tissue was thawed then acclimatized in TissueTek OCT (Ted Pella, Inc., Redding, CA) prior to freezing on dry ice. Sections of DRG (12 μm) and paw tissue (20 μm) were cut with a cryostat (Leica

Biosystems, Buffalo Grove, IL) then thaw-mounted onto Superfrost Plus slides (Ted Pella, Inc.). Sections were dried at room temperature for 2 hr and stored at −20°C prior to staining. Tissue sections were washed 2 × 20 min in PB, incubated with 10% normal donkey serum (RRID:AB_2337258) in PB +0.3% Triton X-100 (PBT) for 1 hr at room temperature, and then incubated with PBT including primary antibodies at 4°C overnight. DRG sections were double-labeled with mouse anti-NeuN (1:250, RRID:AB_2298772) and rabbit anti-ATF3 (1:400, RRID:AB_2258513). Paw sections were double-labeled with mouse anti-PGP9.5 (1:1000, RRID:AB_2572395) and goat anti-collagen Type IV (1:200, Southern Biotechnology, Birmingham, AL). Sections were then washed 3 × 10 min in PB and then incubated with Alexa Fluor 488-conjugated donkey anti-mouse IgG (1:500; RRID:AB_2340850) and Alexa Fluor 594-conjugated donkey anti-rabbit IgG or anti-goat IgG (1:500; RRID:AB_2340621) for 60 min at room temperature. Sections were then washed 3 × 10 min in PB, 2 × 5 min in ddH$_2$O, air-dried, and coverslipped using HardSet VectaShield (RRID:AB_2336787) for imaging. Slides were imaged within 48 hr after mounting followed by storage at 4°C.

## Confocal microscopy

Images of paw biopsies and DRG sections were obtained with a FV1000 laser scanning confocal microscope (Olympus, Waltham, MA) equipped with: Blue diode (405 nm), Argon (458 nm, 488 nm, and 514 nm), Green HeNe (543 nm), and Red Diode (635 nm) lasers using a 20 × 0.75 NA UPlanApo, DIC objective. Image acquisition settings included: 1024 × 1024 resolution, 8-bit image depth, 8.0 µs/pixel scan speed, sequential channel scan with Kalman averaging (2X). Laser power, HV, and offset were adjusted to maximize signal-to-noise ratio and avoid pixel saturation. To facilitate throughput, IENF images for counting were acquired with the pinhole (confocal aperture) set to 650 µm. To maximize image quality, representative IENF images were acquired as image stacks, taken at 0.6 µm optical steps, with two times Kalman averaging, then reconstructed into a 2D z-stack. Adjustment of brightness/contrast, look-up tables, and z-stack reconstructions were done in Fiji (RRID:SCR_002285) (*Collins, 2007*; *Linkert et al., 2010*; *Schindelin et al., 2012*; *Schneider et al., 2012*).

## IENF density quantification

PGP9.5-positive IENFs that crossed the Type IV collagen-stained dermal-epidermal junction into the epidermis were counted in 1–2 randomly selected fields of view (635 µm x 635 µm) per section. Fibers that branched after crossing the dermal-epidermal junction were counted as a single fiber. Fragments of nerve fibers in the epidermis that did not cross dermal-epidermal junction or fibers that approached but did not cross the junction were not counted. Unless junction crossing was indisputably clear, fibers that overlapped with activated Langerhans cells were not counted. The length of the dermal-epidermal junction within each field of view was measured and the total number of fibers that crossed the dermal-epidermal junction were counted to quantify the IENF density (number of IENFs/mm). Quantification was performed using FIJI (RRID:SCR_002285) of at least six sections per paw and the mean was used as the data point for an animal. All counts were conducted by two blinded observers independently to reduce counting bias. Counts were averaged between observers to determine the IENF density for each rat. IENF densities of all rats in each group were used to calculate the mean IENF density ± SEM, then analyzed for statistical significance.

## ATF3 quantification

For analysis of ATF3 expression in DRG sections, background of images was normalized with the FIJI macro BG Subtraction of ROI, followed by despeckling, and binarization. ATF3-positive neurons were defined as having stained nuclei within a size range of 60–300 µm$^2$. To quantify total neurons (NeuN$^+$ cells), images underwent thresholding, applied with a binary Watershed filter, and counted with the Analyze Particles plug-in, where size range was specified as 225 µm$^2$ to infinity (*Obata et al., 2003*). The percentage of ATF3-labeled neurons was calculated by dividing the number of ATF3-positive neurons by the total number of neurons (NeuN$^+$)×100. Quantification was performed using FIJI (RRID:SCR_002285) of at least six sections per DRG per animal. DRG were counted from three animals per treatment group. Values are given as mean ± SEM for statistical comparison.

## Dot blot analysis of PAR accumulation

L4-L6 DRG were dissected quickly, were snap-frozen on dry ice, then stored at −80°C until needed. Lysis buffer (Pierce, ThermoFisherScientific, Waltham, MA), supplemented with 1% phosphatase inhibitor cocktail 3, 100 nM okadaic acid, and 1% protease inhibitor (Pierce, Thermo Scientific) was prepared and stored on ice. Three ganglia (from L4, L5, and L6 DRG) were pooled from each rat and homogenized in 300 μl lysis buffer with a Potter-Elvehjem glass homogenizer. Homogenates were centrifuged at 10,000 x g for 10 min at 4°C. Supernatants were collected and total protein content was determined via protein assay. Samples were diluted in cold lysis buffer to a concentration of 2 μg/μl and stored at −20°C until needed. Dot blot analysis was performed according to the instructions using a Bio-Dot microfiltration apparatus (Bio-Rad, Hercules, CA). Briefly, nitrocellulose membranes were pre-wet in Tris-buffered saline (TBS) and placed on the manifold gasket. The 96-well sample template was aligned over the membrane with the guide pins and secured with vacuum suction. Membranes were rehydrated with TBS, then DRG homogenates were applied in quadruplicate (50 μl per well) and gravity filtered through the membrane for 30 min. Samples were blocked (200 μl of 1:1 TBST and Odyssey Block Buffer (LI–COR Biosciences, Lincoln, NE)) for 60 min on gravity, and then washed twice with TBST on vacuum. 100 μl of monoclonal mouse clone 10 hr anti-PAR (1:300, RRID:AB_2272987) was applied for 45 min on gravity, and then washed three times with TBST on vacuum. Goat anti-mouse IR800 secondary antibody (1:10,000, RRID:AB_10793856) was applied for 45 min on gravity, and then washed two times with TBST on vacuum. The membrane was removed from the manifold, washed twice with TBS, then allowed to dry overnight. Membranes were imaged using a LI–COR Odyssey infrared imager and relative intensities of the innermost dots were quantified using Image Studio (RRID:SCR_013715).

## Cell culture

HeLa cervical cancer cells (RRID:CVCL_0030), Panc-1 prostate cancer cells (RRID:CVCL_0480), MDA-MB-231 breast cancer cells (RRID:CVCL_0062), SK-OV-3 ovarian cancer cells (RRID:CVCL_0532), and SK-N-BE(2) neuroblastoma cancer cells (RRID:CVCL_0528), were purchased from the American Type Culture Collection. HeLa and SK-OV-3 cells were maintained in Basal Medium Eagle with Earle's salts (Sigma) with 10% FBS (Hyclone) and 50 μg/ml gentamicin. Panc-1 cells were maintained in Dulbecco's Modified Eagle's Medium (Gibco, ThermoFisher Scientific) with 10% FBS and 50 μg/ml gentamicin. MDA-MB-231and SK-N-BE(2) cells were maintained in Improved Modified Eagle Medium (Gibco) with 10% FBS and 25 μg/ml gentamicin. Cells were passaged for fewer than 6 months after resuscitation from liquid nitrogen. MDA-MB-231, SK-N-BE(2), SK-OV-3, HeLa, and Panc-1 cell lines were authenticated by STR profiling (Genetica DNA Laboratories, Burlington, NC).

## In vitro NAD/NADH determination

A1A1 cells were derived from retrovirally infected (wildtype SV40 virus) E16 rat cortical neuron cultures (*Berg et al., 1994*). Cells were maintained at 37°C, 5% $CO_2$ in Dulbecco's modified Eagle's medium (DMEM) with 10% FBS. Cells were seeded at a density of 10,000 cells/well in poly-D-lysine-coated, white-walled 96-well plates for luminescence detection. Cells were maintained in serum-free DMEM for 24 hr prior to experimentation. A1A1 cells were then treated with vehicle (ddH$_2$O) or 200 μM $H_2O_2$ for 30 min. Media was aspirated and replaced with DMEM containing vehicle, P7C3-A20 (0.03, 0.3, 3 μM), or NAM (1 mM) for 60 min. FK866 (10 nM) was included with the second treatment where indicated. Following aspiration of media, cells were washed once with PBS, then 50 μl PBS was added to each well. 50 μl of Detection Reagent from the NAD/NADH-Glo Assay kit (Promega, Madison, WI) was added to each well to measure intracellular NAD levels. A1A1 cells were lysed with the Detection Reagent, which includes a NAD cycling enzyme that converts $NAD^+$ to NADH. In the presence of NADH, a reductase catalyzes the formation of luciferin from a proluciferin substrate. The Ultra-Glo recombinant luciferase generates a light signal from luciferin that is proportional to the amount of NAD in the cells tested. White backing tape (Perkin Elmer, Waltham, MA) was attached to the bottom of the plate, followed by luminescence detection (representing intracellular NAD levels) in plate mode with the top optic using a Fluostar Omega Microplate Reader (BMG Labtech, Cary, NC). Relative luminescence units detected per well were normalized to vehicle-treated control wells. For each experiment, treatment conditions were run in

quadruplicate. Data represent mean NAD/NADH levels ± SEM expressed as percentage of vehicle of 4–5 experimental replicates.

## Tissue NAD⁺determination

Hindpaw skin biopsies (3.5 mm punch), sciatic nerve (both sides), and L4-L6 DRG (both sides) were dissected as quickly as possible following decapitation and frozen on dry ice. Approximate times to dissect and freeze paw punches, sciatic nerves, and DRG were 2 min, 8 min, and 12 min, respectively. Samples were kept at −80°C until ready for assay. To determine tissue NAD⁺ levels, samples were transferred from −80°C to a container with dry ice. Tissue samples were weighed frozen (~5–10 mg), transferred immediately into 2 ml of ice-cold homogenization buffer (0.5% dodecyltrimethylammonium bromide, 100 mM $Na_2CO_3$, 20 mM $NaHCO_3$, and PBS; pH 10–11), homogenized with a Tissue Tearor (Biospec) for 15–20 s, then placed on ice. Samples were vortexed every 5 min for 15 min then frozen on dry ice until all samples have been completed. 50 µl aliquots from each sample homogenate were added to a clear 96-well microplate. Standards were prepared in the same manner as the samples. To isolate NAD⁺, 25 µl of 0.4 N HCl was added to the samples, then the plate was incubated at 60°C for 15 min in a dry bath. The plate then cooled at RT for 15 min. 25 µl of 0.5 M Trizma base then was added to the NAD⁺ samples. 50 µl of each sample or standard then was transferred to a 96-well white-walled microplate. 50 µl of the NAD/NADH-Glo kit (Promega) was added to each well, the plate incubated at RT in the dark for 45 min, then was scanned on the Fluostar Omega Microplate Reader (BMG Labtech). Relative luminescence units for each sample were interpolated to determine the NAD⁺ concentration (nmol/l). Protein content (µg/ml) was determined for each sample using the Ionic Detergent Compatibility Reagent (Pierce) with the 660 nm Protein Assay Reagent (Pierce) to normalize tissue NAD⁺ levels (nmol/mg protein).

## Anti-proliferation assay

The sulforhodamine B (SRB) assay was used to determine the effect of P7C3-A20 on the anti-proliferative effects of PTX in various cancer cell lines (*Skehan et al., 1990*). Cells were plated in 96-well plates at a density of 2500–5000 cells/well (depending on the individual growth characteristics of each cell line) and incubated for 24 hr. Cells were treated with P7C3-A20 (0.1–5 µM) or vehicle for 1 hr, and then treated with PTX (0.1 nM – 1 µM). After 48 hr of drug exposure, media was removed and cells were fixed with 10% w/v trichloroacetic acid, washed with $dH_2O$ and then protein stained with SRB dye. After excess dye was removed by washing with 4% acetic acid, the SRB dye was resuspended in Tris and cell density was determined by measuring the absorbance at 560 nm. The inhibition of cell proliferation over the 48 hr of drug incubation was determined for each concentration of PTX ±P7 C3-A20 utilizing a second plate of cells that was fixed at the time of drug treatment as a time 0 measurement, which is represented as a dashed horizontal line at y = 0. Cell density values less than the time 0 measurement indicate cytotoxic activity (*Monks et al., 1991*). The effect of P7C3-A20 (0.1–5 µM) alone on the growth of each cell line also was compared to the growth of vehicle treated cells over the 48 hr period of drug incubation. Nonlinear regression with a four-parameter logistic curve was used to calculate the $IC_{50}$ value for inhibition of cellular proliferation for each independent experiment followed by calculation of the mean $pEC_{50}$ ±SD (n = 3) for each cell line tested.

## Tubulin polymerization assay

The effects of the compounds on purified porcine brain tubulin polymerization (Cytoskeleton, Inc., Denver, CO) were monitored at 340 nm with a SpectraMax plate reader (RRID:SCR_014789). The assay mixture contained 2 mg/ml tubulin in GPEM buffer (80 mM PIPES, pH 6.8; 1 mM $MgCl_2$; and 1 mM EGTA) containing 1 mM GTP and 10% glycerol and DMSO as vehicle (1% v/v) or specified drug in 100 µl reactions at 37°C (*Risinger et al., 2013*).

## In vivo antitumor trial

Female athymic nude *Foxn1^{nu}* mice (Envigo) were maintained in an Association for Assessment and Accreditation of Laboratory Animal Care-approved facility and provided food and water ad libitum. A total of $3 \times 10^6$ MDA-MB-231 cells supplemented with Matrigel were bilaterally injected subcutaneously into each flank. Mice were randomized into treatment groups (n = 5 mice, 8–9 tumors) and

drug treatments initiated when a median tumor volume of 200 mg was reached (~4 weeks). Mice were injected daily with P7C3-A20 (20 mg/kg/day; i.p.) or vehicle. On days 0, 2, and 4, mice also received PTX (11.7 mg/kg; i.p.) or vehicle. Total injection volume never exceeded 0.25 ml. Tumor dimensions were measured with digital calipers on specified days. Tumor volume was calculated using the equation: *mass (mg) = (π/6) × [length (mm) ×width (mm) ×height (mm)]* and the mean change from pretreatment baseline ± SEM was considered for statistical analysis.

## Immunocytochemistry

MDA-MB-231 cells were plated onto glass coverslips and allowed to adhere overnight before compound addition. Cells were pretreated with 5 μM P7C3-A20 or vehicle for 1 hr, then PTX (0.5–100 nM) for 4 hr. After treatment, cells were fixed with methanol (4°C) for 5 min and subsequently incubated with a blocking solution of 10% bovine calf serum in PBS for 20 min at room temperature. Cells were then incubated with a monoclonal β-tubulin antibody (1:400; RRID:AB_1844090) for 2 hr at 37°C. After incubation, cells were washed three times with 1% bovine serum albumin (BSA) in PBS and then incubated with a FITC-conjugated sheep anti-mouse IgG (1:200; RRID:AB_259692) for 1 hr at 37°C. Coverslips were then washed three times with BSA in PBS and stained with 0.1 μg/ml DAPI (Sigma) in PBS for 10 min at room temperature. Coverslips were mounted on slides and visualized with a FV1000 laser scanning confocal microscope (Olympus) using a 60 × 1.42 NA PlanApoN, DIC oil-immersion objective. Adjustments of brightness/contrast and look-up tables were done in Fiji.

## Graphics

All image labels and indicators (i.e. arrowheads), timelines, and schematics were developed with OmniGraffle 6.6.1 (The Omni Group, Seattle, WA). Marvin was used for drawing and displaying chemical structures, Marvin 16.10.10.0, 2016 (ChemAxon, Cambridge, MA (http://www.chemaxon.com)).

## Statistics

Student's t test, one-way ANOVA, or two-way mixed effect ANOVA were used to compare the means among groups, followed by the Dunnett's, Tukey's, or Sidak's post-hoc tests for pairwise comparisons, where appropriate. Pearson correlation coefficients were determined to establish the linear dependence between IENF density and mechanical or cold AUC. Linear regression was used to confirm correlation values and to graph 95% confidence bands of the best-fit line. All statistical tests are two-sided with an alpha of 0.05 as the significance threshold. Analyses were performed in GraphPad Prism 6.0 (RRID:SCR_002798).

## Acknowledgements

We thank AbbVie, Inc. for providing P7C3-A20, P7C3-S321, and A-861696; E Jennings, R Jamshidi, M Pando, J Zamora, L Sullivan, B Jacobs, H Scofield, G Villarreal for technical assistance; R Strong for the dot-blot manifold; S Mooberry for guidance with the cancer cell lines and xenograft studies; M Henry and M Girotti for guidance with the immunofluorescence; D Leippe (Promega) for guidance with NAD metabolite detection; N Williams and L Morlock for LC-MS/MS analysis; and R Cohen, T Esbenshade, K Hunt, A Pieper, J Ready, S McKnight for guidance and fruitful discussions.

## Additional information

### Competing interests

Kelly A Berg, William P Clarke: Received funding to support some of this work in a grant from Calico Life Sciences LLC. The other authors declare that no competing interests exist.

### Funding

| Funder | Grant reference number | Author |
| --- | --- | --- |
| National Center for Advancing Translational Sciences | CTRC Pilot Grant P30 CA054174 | Kelly A Berg William P Clarke |

| National Center for Advancing Translational Sciences | CTSA Pilot Grant 8UL1TR000149 | Kelly A Berg William P Clarke |
| Calico Life Sciences LLC | Research Grant | Kelly A Berg William P Clarke |
| University of Texas Health Science Center at San Antonio | Translational Science Training Program Graduate Student Fellowship | Peter M LoCoco |

The funders had no role in study design, data collection and interpretation, or the decision to submit the work for publication.

## Author contributions

Peter M LoCoco, Conceptualization, Data curation, Formal analysis, Validation, Investigation, Visualization, Methodology, Writing—original draft, Writing—review and editing; April L Risinger, Conceptualization, Investigation, Methodology, Writing—original draft, Writing—review and editing; Hudson R Smith, Teresa S Chavera, Investigation, Methodology, Writing—review and editing; Kelly A Berg, Conceptualization, Resources, Supervision, Funding acquisition, Writing—original draft, Project administration, Writing—review and editing; William P Clarke, Conceptualization, Data curation, Formal analysis, Supervision, Funding acquisition, Validation, Visualization, Methodology, Writing—original draft, Project administration, Writing—review and editing

## Author ORCIDs

Peter M LoCoco http://orcid.org/0000-0002-5678-791X
April L Risinger http://orcid.org/0000-0002-4363-3268
William P Clarke https://orcid.org/0000-0002-8861-8256

## Ethics

Animal experimentation: This study was performed in strict accordance with the recommendations in the Guide for the Care and Use of Laboratory Animals of the National Institutes of Health. The animal study protocol (#20130051AR) was approved by the Institutional Animal Care and Use Committee of the University of Texas Health Science Center at San Antonio and conformed to International Association for the Study of Pain (IASP) and federal guidelines.

## Decision letter and Author response

Decision letter https://doi.org/10.7554/eLife.29626.025
Author response https://doi.org/10.7554/eLife.29626.026

# Additional files

## Supplementary files

• Transparent reporting form
DOI: https://doi.org/10.7554/eLife.29626.023
• Reporting standard 1
DOI: https://doi.org/10.7554/eLife.29626.024

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
