## [Decision Letter]

Thank you for submitting your article "Pharmacological augmentation of NAMPT protects against paclitaxel-induced peripheral neuropathy" for consideration by *eLife*. Your article has been favorably evaluated by a Senior Editor and three reviewers, one of whom is a member of our Board of Reviewing Editors. The following individual involved in review of your submission has agreed to reveal his identity: Clifford J Woolf (Reviewer #2).

The reviewers have discussed the reviews with one another and the Reviewing Editor has drafted this decision to help you prepare a revised submission.

Summary:

This study assesses the ability of the aminopropyl carbazole P7C3-A20 to protect against PTX-induced peripheral neuropathy. The findings show that P7C3-A20 provides robust prevention of PTX-induced behavioral effects of PTX, including mechanical allodynia, and loss of epidermal free nerve endings. The results support the idea that P7C3-A20 acts by enhancing NAD synthesis, although more evidence for this idea is needed. Importantly, P7C3-A20 prevents PTX-induced neuropathy without compromising the anti-tumor efficacy of PTX. The findings suggest that P7C3-A20 co-treatment with the anti-tumor compound PTX may be useful for augmenting NAD synthesis in peripheral neurons thereby preventing chemotherapy-induced peripheral neuropathy. This is an interesting finding that points towards future use of P7C3-A20 or related compounds for prevention of the adverse side effects of PTX.

Concerns raised in review include a lack of mechanistic insight into how NAMPT or P7C3-A20 protects neurons from PTX toxicity, whether the actions of P7C3-A20 are direct on sensory neurons, and the lack of use of a biologically relevant in vitro model system (DRG neurons vs cortical neurons). Moreover, the study does not include standard pharmacological approaches (dose response, pK and pD measurements).

Essential revisions:

1) Measurements of NAD production in DRG neurons following treatment with P7C3-A20 is required to determine whether the drug is acting on the relevant neuronal population. Ideally, this would be done in vivo, however we appreciate that in vivo measurements would be a challenge and potentially complicated due to non-neuronal cells and other limitations, and so in vitro measurements would be minimally required.

2) Does PTX lower NAD levels? Related to point 1, it will be important to do enzymatic assays or measure NAD levels in relevant DRG neurons treated with and without PTX is essential. The section showing that P7C3-A20 rescues NAD levels following treatment with H202 seems extraneous in the absence of any data showing that PTX treatment alters NAD levels.

3) Some dose-dependency and timing of effectiveness of P7C3-A20 for rescue of tactile pain hypersensitivity should be performed.

---

## [Author Response]

Essential revisions:1) Measurements of NAD production in DRG neurons following treatment with P7C3-A20 is required to determine whether the drug is acting on the relevant neuronal population. Ideally, this would be done in vivo, however we appreciate that in vivo measurements would be a challenge and potentially complicated due to non-neuronal cells and other limitations, and so in vitro measurements would be minimally required.2) Does PTX lower NAD levels? Related to point 1, it will be important to do enzymatic assays or measure NAD levels in relevant DRG neurons treated with and without PTX is essential. The section showing that P7C3-A20 rescues NAD levels following treatment with H202 seems extraneous in the absence of any data showing that PTX treatment alters NAD levels.

We thank the reviewers for calling attention to these critical questions. In response to this concern, we conducted another 12-day experimental paradigm to acquire tissue for NAD^+^ analysis. The data have been compiled into a new figure (Figure 7), and we also amended our Discussion points to include these data. Briefly, we found that PTX treatment alone decreased NAD^+^ levels in glabrous hindpaw skin and the sciatic nerve, which was prevented by P7C3-A20. Interestingly, NAD^+^ levels were unchanged in the DRG by either PTX or P7C3-A20. As expected, P7C3-A20 treatment alone did not affect metabolite values. Although this analysis is representative of only a single point in time, it nevertheless supports our hypothesis that stimulation of NAMPT activity with P7C3-A20 protects peripheral sensory neurons from PTX-induced damage.

3) Some dose-dependency and timing of effectiveness of P7C3-A20 for rescue of tactile pain hypersensitivity should be performed.

As we initially set out to assess if P7C3-A20 had any effect on peripheral neuropathy, we optimized the design of the experimental paradigm such that if there were any effect to be seen, we would see it. Accordingly, we began with prophylactic treatment with P7C3-A20 and continued daily injections throughout the 28-day paradigm. We were encouraged to find that P7C3-A20 with this specific treatment schedule reduced behavioral signs of PTX-induced peripheral neuropathy. We kept this prophylactic treatment schedule constant for subsequent studies to pharmacologically characterize P7C3-A20, including dose-response for both the behavioral deficits (i.e., allodynia) and the pathohistologic signatures of nerve damage (i.e., IENF degeneration) induced by PTX treatment. In addition, we used this paradigm to explore mechanistic questions pertaining to NAMPT augmentation by P7C3-A20 and its role in the protection of peripheral neurons from damage.

We agree that evaluation of P7C3-A20 in the context of a treatment/rescue paradigm is an important question, as positive results would have profound implications for treatment for patients with established CIPN. However, rescue requires a different experimental paradigm whereby PTX is administered first to induce damage of peripheral neurons followed by treatment with P7C3-A20. The timing of when to administer P7C3-A20 would be a matter of trial and error and would take quite some time to complete the study. Accordingly, we respectfully suggest that these experiments be a subject for the next manuscript.